# New or Traditional Approaches in Argentina's Bioeconomy? Biomass and Biotechnology Use, Local Embeddedness, and Sustainability Outcomes of Bioeconomic Ventures

**Jochen Dürr** [1,*] and **Marcelo Sili** [2]

1   Center for Development Research (ZEF), University of Bonn, 53113 Bonn, Germany
2   CONICET-Centro de Investigación ADETER, Universidad Nacional del Sur, Bahía Blanca 8000, Argentina
*   Correspondence: jduerr@uni-bonn.de

**Abstract:** The bioeconomy continues to be a contested field in the political debate. There is still no consensus on how a bioeconomy should be designed and anchored in society. Alternative bioeconomy concepts that deviate from the mainstream discourse and are based on small-scale, agro-ecological models are usually underrepresented in the debate. This also applies to Argentina, where the diversity of bioeconomic approaches has not yet been documented and analyzed. The objective of this paper is to identify bioeconomic approaches in Argentina, and characterize alternative, more socio-ecological and locally embedded approaches in order to make them more visible for the political debate. Based on literature research, categories were extracted that can be used to distinguish different types of the bioeconomy. Subsequently, these categories were used in an online survey of 47 enterprises representing different sectors of Argentina's bioeconomy. Using cluster analysis, three groups can be distinguished: a biomass, a biotechnology, and a bioembedded cluster. Argentina's bioeconomy seems to follow a path dependency logic, but new development paths are also opening up. The bioeconomic approaches discovered in Argentina are partly consistent with contemporary bioeconomy typologies, but there is also great diversity within the groups. All bioeconomic approaches have local connections, but are locally embedded in different ways. In addition to the differences between the bioeconomic approaches, two common elements could also be detected: an interest in sustainable use of natural resources and in building networks using synergies with other actors in the territory. These two elements mean that bioeconomic initiatives could pave the way for a new rural development model in Argentina.

**Keywords:** biomass; biotechnology; agro-ecology; territorial development; sustainability; cluster analysis

## 1. Introduction

The importance of locally available resources has again come back into focus as a result of the current crises, and in this context also the bioeconomy, which focuses in particular on a more sustainable and efficient use of local resources [1]. However, the bioeconomy seems to continue a "contested field" [2] in the political debate. Two of the most prominent bioeconomy visions, concepts, and strategies come from the OECD [3], which is strongly biotechnology focused, and from the EU [4], which is more biomass oriented. There are also alternative concepts emerging, such as that from the European Technology Platform TP Organics, which follows a more agro-ecological vision, and stresses the inclusion of different stakeholders from science, politics, business, and civil society [5–7]. There is still no consensus on how a bioeconomy should be designed and how it could be anchored in society. Some authors even argue that these approaches are fundamentally unsuitable for achieving a societal transformation towards a truly sustainable bioeconomy when viewed in the context of global inequalities [8]. In general, alternative bioeconomy concepts

that deviate from the mainstream discourse and are based on small-scale, agro-ecological models are usually underrepresented in the debate and marginalized [6].

This is also the case in Argentina, where the bioeconomy is mainly linked to genetically modified (GM) monoculture crops, intensive use of inputs, and export orientation, with a biotechnological and agro-industrial focus [9]. Argentine agriculture has been driven by the soybean model since 1996, when GM crops were first introduced, and has since expanded greatly in terms of acreage and production levels. This production model induced biotechnology research and innovations, such as drought-tolerant seeds and no-tillage systems [10], but has also had negative consequences on air and water quality, land use changes, land distribution, health and employment [11], and on deforestation [12].

However, official documents of the Argentine government stress the potential of the bioeconomy for regional development, for new industrial developments and local value added, for institutional frameworks, and for sustainable, decentralized, renewable energy supply [13,14]. Yet, as Tittor [14] (p. 325) points out for the case of Argentina: "Agroecological initiatives, a solidarity economy or de-centralized energy systems are not part of the debate." However, the author also mentions that "there are interesting and innovative small-scale projects developing as part of the bioeconomy framework" (p. 324), and that sustainability aspects are gaining in importance in the debate. Moreover, "the agricultural commodity production is still the mainstay of the Argentinian bioeconomy, although small-scale local initiatives, which also include socio-institutional and agro-ecological innovations, are coming up", as Sili and Dürr [15] (p.19) noted. These authors observe the coexistence of two bioeconomic development models, which, following Priefer et al. [16], can be categorized into the technological-based approach, which builds mainly on biotechnologies and genetic engineering for higher biomass production and is oriented towards international markets, and the socio-ecological approach, which envisions a decentralized, ecological agriculture for the development of rural areas through the creation of regional value chains.

This diversity of bioeconomic approaches have not yet been documented and analyzed in Argentina. There is an extensive bibliography at the international level, but only a few studies on the Argentine case, mostly on biotechnological approaches, especially case studies of companies and bioeconomic conglomerates of this sector. For example, on the seed industry that has developed locally adapted soybean seeds [17], on the biotechnology sector and its characteristics, strengths, and weaknesses [18], and on the factors which influence the development of some specific bioeconomic ventures [19]. These studies show that Argentina has caught up in the biotech sector and is now becoming more competitive internationally. Moreover, Argentina has a high biomass potential for developing its bioeconomy [20]. However, there is still no clear description and characterization of a socio-ecological approach in Argentina. Moreover, it is unclear whether and how the bioeconomic approaches can be clearly distinguished from each other in practice, or whether there are also mixed forms.

We have chosen Argentina as a potentially interesting example because, on the one hand, the bioeconomy in this country offers new development opportunities which can overcome the constraints of the previous prevailing models, which were characterized by the contrast between agricultural and industrial development and which can now overcome this in order to contribute to a more balanced territorial development [21]. On the other hand, the private sector has been crucial for the development of the bioeconomy, and only recently have public policies gained importance [22], so bioeconomic models might look different compared to countries where public strategies have been more prominent.

As a hypothesis, we suppose that the socio-ecological approach in Argentina is much less visible, because it has not the size, the capacity for political lobbying, or the same importance for exports, as the technology-based approach. We also hypothesize that there might be combinations of these ideal types of bioeconomic approaches. Despite its low visibility, the socio-ecological approach may have certain characteristics that could make it strategically important for more equitable territorial development, namely: the use of advanced scientific and technological knowledge, together with knowledge based on local

experience, for the generation of dense local linkages, and the possibility of generating local employment and income. Different bioeconomic approaches in rural areas might follow different logics and generate different outcomes for local development, benefiting varying actors, such as small- or large-scale producers. In particular, the socio-ecological approach can be expected to be highly locally embedded, using mainly local resources from low-intensive, small-scaled farming [5,16], and the entrepreneurs to be highly committed to their local communities, a phenomenon that Korsgaard et al. [22] named "entrepreneurship in the rural". To our knowledge, the concept of local embeddedness has not been taken into account in the characterization of bioeconomic approaches so far, which we aim to change in this article.

The main purpose of this paper is to identify the diversity of bioeconomic approaches in Argentina, and characterize the socio-ecological approach, in order to make it more visible for the political debate, which then could also lead to more targeted support policies. So far, to our knowledge, there have been, no attempts to group bioeconomic enterprises into the different ideal types of the bioeconomy described in the literature. Based on some recent literature that categorize such bioeconomic approaches, we extracted categories that can be used to distinguish different bioeconomy types. Then, we used these categories in an online survey with 47 bioeconomic enterprises representing different sectors of the Argentine bioeconomy. The novelty of this approach lies in the operationalization of categories that characterize bioeconomic types and their application to the real business world. In this way, we aim to answer three important research questions: First, can different bioeconomic approaches be clearly distinguished in the case of Argentina? Second, what are the characteristics of the bioeconomic approaches that could be identified? Third, what kind of linkages do these approaches maintain with rural territories, and what impact do they have on sustainability there?

The paper is structured as follows: First, we describe the main bioeconomy typologies discussed in the literature, and the concept of local embeddedness of enterprises. Second, we explain the methods used. Third, we describe the three bioeconomic groups which emerged from the cluster analysis. Fourth, we discuss the bioeconomic approaches and their linkage with territorial development. Finally, we provide recommendations for further research on the bioeconomy in Argentina.

## 2. Bioeconomy Typologies and Local Embeddedness

### 2.1. Bioeconomy Typologies

Bugge et al. [5] identified three ideal types of the bioeconomy: (1) a biotechnology vision, (2) a bio-resource vision, and (3) bio-ecology vision. These visions can be characterized by some key variables: aims & objectives, value creation, drivers & mediators of innovation, and the spatial focus. The aims and objectives of vision 1 are mainly focused on economic growth and job creation. This also applies to vision 2, where sustainability also plays a central role. Aspects such as sustainability, biodiversity, conservation of ecosystems, and avoiding soil degradation are crucial for vision 3. The value creation of vision 1 is based on biotechnologies, and the commercialisation of research & technology, while vision 2 focuses on the conversion and upgrading of bio-resources, and vision 3 on the development of integrated production systems and high-quality products with territorial identity. The innovation process of vision 1 follows a linear model of transforming biotechnological research into new products and processes, stresses the importance of cooperation with universities and research centers, and is based on patents, whereas the innovation drivers of vision 2 relate to optimizing the use of land, bio-resources, and waste, and is more interdisciplinary and network-oriented. Vision 3 drivers are based on the search for sustainable agro-ecological practices, re-use and recycling of waste, and efficiency in land use, and research and innovation activities are related to transdisciplinary sustainability issues. Finally, the spatial focus of vision 1 is on global biotechnology centers and regions, while vision 2 highlights the potentials for rural development, also in peripheral regions,

as does vision 3, but with a stronger emphasis on the development of territorial identities and locally embedded economies.

The scientific and social debate on the bioeconomy was analyzed by Priefer et al. [16]. Using several key categories, the authors identified two different approaches, which they call the technology-based vs. the socio-ecological approach. The technology-based approach stresses the importance of increasing biomass production through intensification of agriculture, but increasingly also through laboratories, and the major role played in this process by biotechnologies and patents, multinational companies and global value chains, international competitiveness and innovations, centralized solutions and economies of scale, the partnership between politics, science, and companies, as well as the promotion of life sciences. In contrast, the socio-ecological approach emphasizes multifunctional, ecological agriculture, natural cycles and reduced resource consumption, the promotion of social innovations, the use of local knowledge, the strengthening of rural areas, the creation of regional value chains, a more localized food and energy supply based on small-scale, region-specific biomass production with greater participation by civil society, and inter- and transdisciplinary research. According to the authors, these approaches are not necessarily mutually exclusive, and some features might be connected or otherwise combined.

By doing literature research, expert interviews, and conference participations, Vivien et al. [7] characterized three ideal-types of bioeconomy narratives that revolve around the notions of socio-technical relations, governance, sustainability, and tensions/paradoxes. Bioeconomy type I, based on the works of Georgescu-Roegen, is defined as an ecological economy, respecting the limits of the biosphere, whereas bioeconomy type II is defined as a biotechnology based economy driven by science, and type III as a bio-based economy that replaces fossil fuels with biomass. Type I takes a strong sustainability approach, promotes an economy of prudence and sharing, and favors democratic, ecological planning, while criticizing pure technical solutions. Type II, on the other hand, sees the techno-scientific promises of the bioeconomy, is based on commodification of knowledge (patents), but has a weak sustainability approach. Bio-refineries are at the heart of bioeconomy type III, which pursues mission-driven policies to identify ecological transitions, by substituting products and processes. However, this is still done within a weak sustainability concept, and with the problem that pressure on resources and land could increase.

Through discourse analysis of official policy documents and stakeholder interviews, Hausknost et al. [6] propose two dimensions by which the different visions of the bioeconomy can be located in a continuum: on the one hand the technological dimension, ranging from visions of agroecology to industrial biotechnology; and the political–economic dimension ranging from notions of sufficiency to capitalist growth. This allows the authors to distinguish four different areas, namely (1) Sustainable Capital, founded on the belief of bio-technologies and industrial innovations for further, sustained and sustainable economic growth; (2) Eco-Growth, based on the narrative of agro-ecological innovations for intensification and efficiency gains; (3) Eco-Retreat, characterized by the combination of ecological practices and socio-economic sufficiency; and (4) Planned Transition, constituted by a high-tech vision together with a sufficiency approach. Different actors (state, business, academia, civil society) not only have different visions, but also their roles for the transition to the bioeconomy need to be critically assessed.

In analyzing the European bioeconomy agenda, Levidow et al. [23] describe what they call the (dominant) life science agenda versus the (marginalized) agro-ecological agenda. The former aims to modify plants and animals for greater productivity or new uses, and to convert biomass into various inputs and outputs that can be de- and recomposed for different industrial products, and relies on laboratory knowledge and bio-refinery plants. The latter looks for agro-ecological systems that minimize the use of external inputs, emphasize product identity with territorial characteristics that can be recognized by consumers and therefore add local value, and is based on small-scale farming units and knowledge of agro-ecological methods.

Although this is a short literature review, it presents key issues in the current discussion on the bioeconomy. The typologies elaborated by the authors cited, referred to as bioeconomic "visions", "approaches", "agendas", "narratives", or "models", reflect official documents, research publications, public discourses, etc. Yet, these typologies have been developed using different categories that are not necessarily compatible with each other. Moreover, none of the authors listed above analyzed whether in the real world of bioeconomic ventures, the different bioeconomy types can be found and clearly distinguished. Furthermore, the territorial dimension of the bioeconomy is mentioned by some authors, but not described in detail. If the bioeconomy is to foster territorial development, it seems crucial that the territorial dimension of the bioeconomy is considered more thoroughly (see the following section). As our aim was to apply the concepts to business companies, we extracted three main dimensions, which were then further used for the development of variables that could be utilized in a cluster analysis to characterize bioeconomic approaches (see Section 3.2): (1) biomass production and use; (2) technology, research, and innovations; and (3) sustainability impacts and territorial linkages. Table 1 shows in highly summarized form how the authors describe the main characteristics of the different approaches in relation to (1)–(3).

**Table 1.** Characteristics of different typologies of the bioeconomy.

| | **Biotechnological Approach** <————————————————————> **Socio-Ecological Approach** | | | |
|---|---|---|---|---|
| Bugge et al. [5] | Biotechnology Vision (1) biomass transformation into marketable products (2) biotechnologies based on R&D (3) global markets | Bio-Resource Vision (1) upgrading bio-resources and optimizing land use and waste (2) engineering and science (3) rural development, but weak sustainability | | Bio-Ecology Vision (1) sustainable agro-ecological practices (2) transdisciplinary (3) territorial identity, strong sustainability |
| Hausknost et al. [6] | Sustainable Capital (1) eco-efficient use of renewable resources (2) biotechnologies and industrial innovations (3) global economic growth | Planned transition (1) reduced resource use (2) high biotech vision (3) sufficiency approach, global trade | Eco-Growth (1) organic farming (2) agro-ecological innovations (3) regional, small-scale | Eco-retreat (1) ecological practices (2) small-scale, democratic control over technologies (3) socio-economic sufficiency |
| Vivien et al. [7] | Science-based economy (1) industrial biotechnologies, cell factories (2) commodification of knowledge, patents (3) weak sustainability | Bio-based economy (1) replacing fossil resources by biomass (2) heterogeneous knowledge base (3) weak sustainability | | Ecological economy (1) respecting the limits of the biosphere (2) prudence, against "promethean technologies" (3) strong sustainability |
| Priefer et al. [16] | Technology Based-Approach (1) intensive production, efficiency gains (2) biotechnologies, competitiveness, technology leadership, patents (3) multinational companies and global value chains | | Socio-Ecological Approach (1) multifunctional, ecological agriculture, reduced resource consumption (2) social innovations, local knowledge, transdisciplinary research (3) regional value chains, autarchy, local stakeholders | |
| Levidow et al. [23] | Life science trajectories (1) modifying plants and animals, conversion of biomass (2) lab knowledge and bio-refineries (3) competition in global markets | | Agro-ecological trajectories (1) systems that minimize external input use (2) knowledge systems for agroecology (3) territorial identity and distribution systems | |

Source: own elaboration.

Different visions translate into various paths to follow for a transition to the envisioned bioeconomy: for example, a technology-based transition based on research and on techno-

logical innovations that can be transformed into competitive bio-based products, opposed to a socio-ecological transition that follows sustainability concerns of the biosphere's supply and regeneration capacities. However, there is also a need and possibility to integrate the different perspectives of the transition [24,25]. The bioeconomic transition pathways proposed by Dietz et al. [26] are not so much influenced by visions, but take into account the different roles that techno-economic mechanisms can play in the development of the bioeconomy, especially for factor substitutions and for efficiency gains, and differentiate the (1) fossil-fuel substitution, (2) primary sector productivity enhancement, (3) new and more efficient biomass uses, and (4) low-bulk and high value applications pathway. Sili and Dürr [15] proposed, for the case of Argentina, a fifth pathway defined as the "generation of new, innovative products and services with local value added."

A value chain approach to differentiate bioeconomic types and a potential upgrading process is proposed by Mac Clay and Sellare [27]. The authors distinguish six value chain models with different characteristics of biomass and biotechnology use and innovation processes. The concept of bioeconomic upgrading from high-volume, low value to low-volume, high value chains describes possible trajectories towards lower environmental impacts with higher economic opportunities. Finally, Bröring et al. [28] differentiate four innovation types (IT) in the bioeconomy, namely (I) substitute products, (II) new processes, (III) new products, and (IV) new behavior. These innovation types are associated with specific challenges, related to markets, value chains, resources, innovation capacities, consumers, and sustainability. For example, for IT I, the integration of bio-based substitutes into existing value chains is a particular challenge, and there is market competition with the fossil-based industry. For IT II, technology adoption and diffusion as well as knowledge transfer are important challenges for establishing new processes and value chains. One of the biggest challenges of IT III is innovation capacity, as new products require long development periods, high investments, and the transfer of biotechnologies to users. Finally, for IT IV, behavioral changes are confronted with different problems such as lack of acceptance (of new products or processes), knowledge gaps, insufficient public communication, and unwillingness to change. The typology could be used to analyze the dynamics and impact of policies and other factors on innovations in the bioeconomy, as well as their sustainability performance [28].

### 2.2. Local Embeddedness

One of the advantages of the bioeconomy is seen in the promotion of rural development and stimulation of local economic and social development, with the possibility to adapt to local characteristics and based on knowledge of local stakeholders [29]. Especially the agro-ecological version of the bioeconomy is associated in the literature with local knowledge, small-scale production units, shorter supply chains, territorial identity, and rural development [30].

Bioeconomic ventures in rural areas may follow different logics and therefore achieve different outcomes for local development. One important differentiation in this respect might be what Korsgaard et al. [22] idealized as "entrepreneurship in the rural" vs. "rural entrepreneurship". The authors argue that the former is weakly embedded in the rural space, i.e., follows a profit-oriented choice of location, which can also vary depending on the (economic) framework conditions. Entrepreneurs in the rural react to economic incentives (such as low land prices or labor costs) in their location decision, and the location is considered as a space for profit-making, but they are not very engaged in local communities and have no specific interest in rural development (which is not to say that these ventures cannot have positive development impacts). In contrast, the latter represents activities that rely on specific local resources that cannot be easily exchanged, and involves a particular commitment of the company to its location and is heavily place-related, which entails a different logic regarding the location decision and makes a spatial relocation more difficult or unlikely. Rural entrepreneurs see the location also as a space for social life, and they create new values with local resources that contribute to the development of

the communities from which these resources originate, not only in economic, but also in socio-cultural terms [22].

Rural entrepreneurial processes are influenced by their spatial context such as the resource base and market outlets, which also determine whether these processes are more or less embedded locally. On the one hand, this depends on the resource endowment and the extent to which rural enterprises use local resources. On the other hand, embeddedness also hinges on whether and how rural entrepreneurs connect their local place to non-local spaces, i.e., whether they link (or not) the local to the national or global economy. "Bridging" of localized resources and products to non-local spaces, i.e., market outlets outside the territory, can lead to dynamics and opportunities for the local economy [31]. This also means that some enterprises might be locally embedded through their resource base, but not through their customers (which are mainly non-local), and vice versa, leading to four types of rural entrepreneurs and their embeddedness, depending on the extent to which local resources are used and the extent to which there is "bridging" to non-local costumers: (1) low embeddedness, but high bridging of local resources to non-local customers, (2) high embeddedness and high bridging to non-local customers, (3) high embeddedness and low bridging to non-local customers, and (4) low embeddedness and low bridging.

This means that enterprises with high local embeddedness might share many characteristics of the socio-ecological approaches (see Table 1), for example, minimizing the use of external inputs and showing local identity, which are mentioned by Bugge et al. [5] and Levidow et al. [23]. The importance of regional value chains, autarchy, and the connection to local stakeholders [16] also aims in this direction. However, by bridging local resources to non-local costumers, there might also be cases that belong more to the approach of biomass transformation into marketable products [5] by replacing fossil resources by biomass [7], and cases where enterprises are active on international markets, a characteristic rather belonging to the biotech model [16]. We have therefore included local embeddedness as an additional category to characterize bioeconomic approaches.

## 3. Methods

### 3.1. Selection of Cases and Sample Structure

In the absence of consolidated databases in Argentina, which might be, inter alia, due to lack of a clear definition of the bioeconomy, and thus a clear delineation of which enterprises "belong" to the bioeconomy and which do not, we took an ad hoc approach using three of the existing, but surely incomplete, lists of bioeconomic ventures. First, we consulted the list of the Ministry of Science and Technology and the Ministry of Agriculture of Argentina, which comprises 110 ventures linked to the bioeconomy. Second, we used the list of the web portal "Bioeconomy in Argentina", developed by Argentine agricultural journalists, which includes 30 ventures. Thirdly, we utilized a list of 20 companies interviewed in the documentary on the Argentine bioeconomy, produced by public television. In total, a list of 160 ventures was compiled, but due to the high number of repetitions between the different lists, a final list of 102 cases was consolidated.

Since we do not know the criteria used to create the lists, nothing can be said about possible selection biases. Nevertheless, the companies included are highly diverse in terms of sectors, regions, scale, technological level, etc. They also seem to belong to different bioeconomic approaches (shown in Table 1). For example, there are companies from the biotechnology sector, there are biofuel enterprises, and there are organic producers. Our aim was to include as many as possible of the listed firms, so we tried to contact all of the 102 firms by sending emails and, if no reaction occurred, by reminding them with a telephone call. In the end, 48 of the 102 contacted enterprises filled out the questionnaire (response rate of 47%). One of the enterprises had to be eliminated because it turned out to be a recycling of electronic devices firm, resulting in the following sample structure of 47 enterprises (see Table 2). The distribution of the sectors as well as the regions and the size of the companies (measured by the number of employees) is relatively even. However,

the Cuyo region is missing. The map in Appendix C shows the location of the 47 enterprises in the different regions of Argentina.

**Table 2.** Structure of the sample.

| Sector | No. of Cases | Region | No. of Cases | Size | No. of Cases |
|---|---|---|---|---|---|
| Food and Beverages | 10 | Metropolitan | 8 | Very small | 10 |
| Bioenergy | 12 | Pampa | 18 | Small | 13 |
| Agro-Inputs | 11 | Patagonia | 5 | Medium | 9 |
| Pharma and Cosmetics | 6 | Northeast | 8 | Big | 9 |
| Biomaterials | 8 | Northwest | 8 | Very big | 6 |

*3.2. Data Collection*

In order to operationalize the different types of the bioeconomy described above, our aim was to distill the main categories from the literature, find proper variables for each category, and formulate appropriate questions suitable for the use of a Likert-type scale. As mentioned above, we extracted three main topics with opposing views from literature, which refer to (1) biomass production and use (intensive agriculture vs. agro-ecological systems; large scale use of biotechnologies and biomass production vs. small scale, circular systems); (2) technology, research and innovations (technology-based vs. socio-ecology-based; life sciences, R&D and patent-based vs. transdisciplinary and agro-ecological practices-based), and (3) sustainability issues and territorial linkages (weak vs. strong sustainability; high input monocultures vs. conservation of ecosystems, biodiversity and soils; global vs. regional value chains; central regions vs. peripheral regions; international players vs. local stakeholders). Furthermore, we added from the literature of local embeddedness the resource base (local/non-local input and outlet markets), locational choices (a-spatialized vs. sustainable place making), cooperation with local stakeholders, and the role local identity plays for the entrepreneurs.

We grouped these topics into four categories: biomass, scale, technology, and territoriality, and used 19 variables to describe them, see Table 3. The exact questions are listed in the questionnaire in Appendix B. The variables are mainly ordinal (such as "very small" to "very high" or "0%", "1–24%", etc. to "100%"), so that a Likert-type scale could be used. We decided to use a 5-point scale as a sufficient, not too differentiated scale. Score "1" is supposed to be closest to the socio-ecological approach, and score "5" would fully represent the biotech–biomass focused approach, with the exception of the category "biomass", where low values are also to be expected for the biotech approach. Taking the category "biomass" as an example, score "1" stands for low volumes of biomass used, produced by small-scale farms with no intensive production methods, and score "5" would be a highly intensive production of high biomass volumes by large farms. "Size" was measured by three variables (turnover, number of employees, and production volumes compared to other companies). Again, low levels are to be expected for the socio-ecological approach. In the category "technology", the percentages show how much of total production value depends on biotechnologies, local knowledge, and patents. We wanted to know how much companies cooperate with scientific and with private organizations. The socio-ecological approach was expected to fall into low levels of all variables except for the variable "local knowledge". Hence, for this variable, low scores mean high importance. In the category "territoriality", we asked which markets are mainly served, from where inputs mainly come from (excluding biomass, which was asked in the "biomass" category), and how much international prices influence the profitability of the business. Moreover, we wanted to know to what extent products are based on local identity, to what extent business activities contribute to an improved environment and to a more sustainable use of natural resources, and to what extent the company interacts with local stakeholders. Note that some variables are inversely formulated, i.e., the higher the variable outcome, the lower the score. This occurs mainly in the category "territoriality", meaning that higher local embeddedness

and higher contribution to the environment are associated with lower scores, which is expected for the socio-ecological approach. Three of the variables are nominal (Biomass 2, Territoriality 1 and 2, which go from "local" to "global"), so that no scale points were used.

**Table 3.** Variables used to differentiate bioeconomic approaches.

| Categories | Scale | | | | |
|---|---|---|---|---|---|
| Biomass | 1 | 2 | 3 | 4 | 5 |
| 1: Volumes | Zero | <10 t | 10–100 t | 100–1000 t | >1000 t |
| 2: Origin * | Local | Regional | National | L. America | Global |
| 3: Production Scale | Very small | Small | Medium | High | Very high |
| 4: Intensity | No use | Low | Medium | High | Very high |
| Size | 1 | 2 | 3 | 4 | 5 |
| 1: Turnover (1000$) ** | <50 | 50–250 | 250–1000 | 1000–10,000 | >10,000 |
| 2: Compared to others | Much smaller | Smaller | Average | Bigger | Much bigger |
| 3: No. of Employees | 1–5 | 6–20 | 21–100 | 101–500 | >500 |
| Technologies | 1 | 2 | 3 | 4 | 5 |
| 1: Biotechnologies | 0% | 1–24% | 25–49% | 50–74% | 75–100% |
| 2: Local knowledge | 75–100% | 50–74% | 25–49% | 1–24% | 0% |
| 3: Patents | 0% | 1–24% | 25–49% | 50–74% | 75–100% |
| 4: Scientific coop. | Not | Not Much | Medium | Much | Very much |
| 5: Private sector coop. | Not | Not Much | Medium | Much | Very much |
| Territoriality | 1 | 2 | 3 | 4 | 5 |
| 1: Markets * | Local | Regional | National | L. America | Global |
| 2: Suppliers * | Local | Regional | National | L. America | Global |
| 3: International prices | No influence | Some inf. | Medium inf. | High inf. | Very high inf. |
| 4: Local identity | Very much | Much | Medium | Not much | Not |
| 5: Environment | Very much | Much | Medium | Not much | Not |
| 6: Natural resources | Very much | Much | Medium | Not much | Not |
| 7: Local stakeholders | Very much | Much | Medium | Not much | Not |

* no scale used; ** Argentine Pesos (ARS); at the time of the interviews, ARS 100 = USD 1.

In addition to the 19 variables shown in Table 3, an open-ended question on the type of products and services was asked, as well as multiple choice questions with predetermined answers on four key characteristics of the firms, three reasons of their locational choice, three levels of product specialization, and the type(s) of pathways followed (see the questionnaire).

The survey was conducted as an online survey between October and December 2021. An online survey with a standardized questionnaire was considered to be an appropriate tool, firstly to do justice to the ongoing epidemic and secondly to provide the ready-made scales. The answers were formulated in a Likert-type form, for which Google forms was used. The interview questionnaire was developed in Spanish; an English translation is added in Appendix B.

*3.3. Data Analysis*

In a first analytical step, we divided the highly diverse 47 enterprises by their description of activities, complemented by information given in their websites, into five sectors: (1) Bioenergy, (2) Biomaterials, (3) Food and beverages, (4) Cosmetics and pharmaceutics, (5) Agro-inputs, see Table 2.

A hierarchical cluster analysis (Ward method, using quadratic Euclid distance) was carried out with the values of the ordinal variables. However, we had to exclude two of these variables (Size 1 and Size 2) due to missing data. We decided to exclude the two variables rather than the (three) cases so as not to reduce the total number cases, which were already few. Moreover, the variable Size 1 ("turnover") was biased (64%) towards the upper scale point of "5" (over ARS 10 million), the cut-off of which had been chosen too low, so that the variable was not able to differentiate the enterprises well. In addition, there were four answers of "zero biomass used", so that the following questions on origin, scale, and intensity of biomass use had be automatically scored as "1", even if there was no biomass production at all.

The use of cluster analysis was intended to answer the question of whether there are relatively homogenous groups, as expected the biomass, the biotech, and an agro-ecological group, or whether there are more distinguishable groups with different characteristics, or whether there are no clearly distinguishable groups at all. This could lead to a more differentiated typology, for example, small-scale biotech enterprises with high local embeddedness, or large-scale, international biomass-related enterprises with agro-ecological characteristics, etc.

A first analysis using 14 variables showed no preferred number of clusters. With a pre-defined number of clusters (3), results showed highly diverse groups of enterprises where no clear pattern could be detected. We then decided to concentrate on only four variables that were considered decisive for the three approaches, namely volumes of biomass, size, use of biotechnologies, and cooperation with local stakeholders. The result for three clusters showed distinguishable groups, but with some enterprises belonging to the "wrong" cluster (two clearly biotech-focused and one clearly biomass-focused companies).

We then decided to use a very simple method, guided by the underlying knowledge and theory of the different types of production models: Firstly, the cluster of enterprises which probably would belong to the biomass-based approach was defined by separating all companies with high and very high biomass utilization volumes (scale 4 and 5), resulting in a group of 21 companies, mainly from the bioenergy and food sectors. Secondly, the biotechnology group was defined as all companies that mainly base their productive processes on biotechnologies (level 5 and 4) plus the companies with medium levels of biotechnology use (level 3) if cooperation with scientific and technological organizations has some importance for these companies (at least level 2). There was an overlapping of six enterprises with the biomass approach that were excluded from the biotech group, resulting in 15 companies, mostly active in pharmaceuticals and agro-inputs. The third bioeconomic group was elaborated as the residual of all companies not belonging to the already defined biomass and biotechnology groups, resulting in 11 companies belonging to different sectors, in particular bio-products and foodstuffs. The result of this simple algorithm was similar to the cluster analysis with four variables, but with the difference that the three biotech companies now belonged to the "right" group. We therefore decided to use the three clusters produced by the simple algorithm, using only two variables (biomass and biotechnology use).

Each of the three clusters was characterized by the number of enterprises belonging to each of the five scale levels of each variable. It was also determined how the other variables not used for cluster analysis were distributed between the clusters. For each variable, analysis of independence was carried out, using Fisher's exact test statistics. A non-parametric correlation analysis (using Spearman's Rho) was performed with the scale points of all ordinary variables for each cluster separately. In this way, it can be investigated whether there are strong and significant relationships between certain variables of the

different clusters. For some of the variables that showed such relationships, a second cluster analysis was then taken for each of the three clusters (using Ward method) to detect sub-groups in each cluster.

## 4. Results

Derived from the literature and the cluster analysis, we have identified three bioeconomy approaches in our material: the biomass, the biotechnological and an alternative cluster we will describe below in further detail. For this cluster, according to the main characteristics of this group, and deviating from the names used in literature so far, we propose an alternative name, i.e., the locally embedded bioeconomy approach, in short, the bioembedded approach.

Table 4 presents a synthesis of each of these clusters according to the different variables of analysis. To differentiate more clearly, only the most frequent scale levels are shown (if there are equally frequent levels, all of them are listed), and used the same colors as in Figure A1 of the Appendix A, where all levels are presented. Dark green stands for the lowest level 1, dark red for the highest level 5, and medium level 3 is painted in yellow. The differences were tested with Fisher–Freeman–Halton exact statistic, and are significant for biomass 1, biomass 3, size, technology 1, and territorial 2.

**Table 4.** Cluster characteristics of scale variables.

| Variables | Fisher Exact | Cluster 1. Biomass (n = 21) | Cluster 2. Biotechnology (n = 15) | Cluster 3. Bioembedded (n = 11) |
|---|---|---|---|---|
| Biomass 1: Volume | 51.3 ** (0.000) | >1000 tn: 71% | <10 tn: 73% | <10 tn: 82% |
| Biomass 2: Origin | 6.3 (0.346) | local: 71% | local: 53% | local: 82% |
| Biomass 3: Scale | 15.9 * (0.025) | medium: 48% very high: 29% | small, very small, medium: 27% | very small: 64% |
| Biomass 4: Intensity | 12.6 (0.092) | medium: 38% low: 24% | no use: 47% | no use: 36% low: 36% |
| Size 3: No. of Employees | 19.5 ** (0.005) | 101–500: 33% >500: 24% | 1–5: 33% 6–20: 33% | 1–5: 45% 6–20: 27% |
| Technology 1: Biotechnologies | 28.4 ** (0.000) | level 2: 43% level 1, 3: 19% | level 5: 60% | level 2: 64% |
| Technology 2: Local knowledge | 8.8 (0.324) | level 2: 38% level 3: 29% | level 1: 33% level 4: 27% | level 3: 45% level 1, 2: 18% |
| Technology 3: Patents | 7.5 (0.490) | level 1: 57% | level 1: 60% | level 1: 45% level 2: 27% |
| Technology 4: Scient. Cooperation | 4.7 (0.848) | level 4: 38% level 3: 29% | level 4: 33% level 2: 20% | level 3: 27% level 4: 27% |
| Technology 5: Private Cooperation | 6.1 (0.682) | level 4: 38% level 2, 3, 5: 19% | 33% level 1 27% level 4 | 45% level 4 18% level 3,5 |
| Territorial 1: Markets | 5.3 (0.486) | national: 48% international: 29% | national: 53% international: 27% | national: 73% international: 27% |
| Territorial 2: Suppliers | 12.9 * (0.022) | national: 76% local: 14% | international: 40% national: 40% | national: 36% international: 27% |
| Territorial 3: Internat. Prices Influence | 3.5 (0.790) | very high: 33% high: 29% | high: 47% medium: 33% | medium: 36% high, very high: 27% |
| Territorial 4: Local Identity | 3.3 (0.986) | much: 33% very much: 29% | medium: 33% much: 33% | very much: 36% much: 36% |
| Territorial 5: Environment | 8.7 (0.341) | very much: 38% much: 24% | much: 53% | much: 36% very much, medium: 27% |
| Territorial 6: Natural resources | 8.9 (0.304) | much: 43% very much: 38% | much: 33% not much: 27% | very much: 36% much: 27% |
| Territorial 7: Local Stakeholders | 11.5 (0.115) | much: 52% very much: 33% | much: 47% medium, not much: 20% | much: 45% not much: 27% |

* significant ($p < 0.05$) ** significant ($p < 0.01$).

Cluster 1 uses in 71% of the cases more than 1000 t of biomass, whereas in the other two clusters, most enterprises use less than 10 t per year. The scale of biomass production is also medium or very high in Cluster 1, while it is mostly very small in Cluster 3. Cluster 1 consists of mainly larger companies with more than 100 or even 500 employees, while the other two clusters mostly employ only up to 20 people. Of Cluster 2, 60% heavily use biotechnologies, while in Cluster 3 most enterprises (64%) use biotechnologies only to a small extent. Finally, input suppliers (excluding biomass) are overwhelmingly (76%) national in Cluster 1, while in Cluster 2 and 3 the picture is more mixed, including international suppliers. It seems that the equipment necessary for Cluster 1 enterprises can already be produced by national suppliers (provided that they do not import them), while some specialized inputs needed for Cluster 2 and 3 still require imports.

Even if for the other variables no significant relationships with the clusters could be detected, Table 4 demonstrates that the most frequent scale levels differentiate between the clusters. For example, 82% of enterprises of Cluster 3 source their biomass locally, which is only the case for 53% of Cluster 2 enterprises. Scientific cooperation interestingly is higher for Cluster 1 than for Cluster 2, which could mean that biomass-related enterprises nowadays are searching for new knowledge and innovation capacities, while biotech companies might be more independent from public R&D institutions, as they often possess their own laboratories and research departments. Also interestingly, patent use is more common in Cluster 3, where only 45% of enterprises do not use patents, compared to 57% and 60% of Cluster 1 and 3, respectively. Cooperation with the private sector is stronger in Cluster 1 and 3, where 38% and 45% of enterprises fall in level 4, which is only the case for 27% of Cluster 2. 73% of enterprises in Cluster 3 serve the national market, whereas only around half of Cluster 1 and 3 enterprises do this. Local identity is slightly more important for Cluster 3 enterprises (72% high or very high, in contrast to 62% and 60%, respectively, of Cluster 1 and 2). Only 9% of Cluster 3 enterprises do not contribute positively to the local environment, whilst 27% of Cluster 2 and 20% of Cluster 1 stated this. Also, more enterprises (27%) of Cluster 2 contribute only little to sustainable use of natural resources, probably because many of these enterprises do not use any biomass at all. Finally, cooperation with local stakeholders is most common for Cluster 1.

Another feature can be detected in Table 4, and even better in Figure A1 (see Appendix A): there are some intra-cluster differences, which might make subgrouping worthwhile. For example, in Cluster 1, big companies predominate, but there are also some small businesses (24%) involved. Around half of enterprises of Cluster 2 do use local, traditional knowledge, the other half, not (much). Of Cluster 3, 45% cooperate much with local stakeholders, 27% little, etc.

Apart from some variables, clusters are also not independent from the sector they belong to, the pathway they follow, or the type of products they produce, see Table 5: the bioenergy sector predominates (57%) in Cluster 1, whereas in Cluster 2 the agro-input (40%) and the pharmaceutical sector (33%) stand out, and in Cluster 3, the biomaterial and the food sector are, each with 36%, most important. In Cluster 1, 57% of the enterprises follow pathway 1, substituting fossil resources, whereas Cluster 2 enterprises tread pathways 2 and 4, the productivity enhancement and the low volume–high value pathway, and 45% of Cluster 3 take pathway 5, creating new, innovative products with local value added. Finally, Cluster 1 is mainly producing standardized (43%) and specialized products (47%), while Cluster 2 (53% and 47%, respectively) and 3 (45% and 36%, respectively) concentrate on niche and specialized products.

**Table 5.** Cluster characteristics of qualitative variables.

| Variables | Fisher Exact | Cluster 1. Biomass (n = 21) | Cluster 2. Biotechnology (n = 15) | Cluster 3. Bioembedded (n = 11) |
|---|---|---|---|---|
| Regions | 11.4 (0.181) | Pampa 48% NOA 24% | Pampa 48% Metropolitana 24% | Patagonia 27% |
| Pathways | 23.7 ** (0.000) | P1 57% | P2 40% P4 40% | P5 45% P1, P2 18% |
| Product Characteristics | 14.1 ** (0.005) | Specialized 48% Standardized 43% | Niche 53% Specialized 47% | Niche 45% Specialized 36% |
| Sectors | 30.5 ** (0.000) | Bioenergy 57% Food 23% Agro-inputs 14% | Agro-inputs 40% Pharmacy 33% Biomaterials 20% | Food 36% Biomaterials 36% Agro-inputs 18% |
| Keywords | no results possible | Local development 18% Sustainability 15% Add value 15% | Innovation 20% R&D 20% Biotechnology 15% | Add value 22% Sustainability 20% Innovation 15% |
| Location criteria | 17.9 (0.530) | Raw materials 43% Local connections 21% Social, cultural, and environmental quality 10% | Local connections 24% Social, cultural, and environmental quality 14% Raw materials, skilled labor 14% | Social, cultural, and environmental quality 25% Local connections 21% Raw materials 17% |

** significant ($p < 0.01$).

### 4.1. Cluster 1. Biomass Approach

The companies that make up this cluster are characterized by the use of large quantities of biomass (71% of them use more than 1000 tons), which, due to the high cost of its mobility, comes from the same areas (71%). These companies often integrate primary production with the transformation of biomass into bioenergy or foodstuffs for human consumption, but more especially for animal consumption. In short, these are generally companies that have historically developed their activities in primary and food production (soya, maize, meat, sugar), but which have been able to move up the value chain, taking advantage of their experience in the primary and processing sector, the available infrastructure and equipment, and the scientific and technological facilities in their regions. The biomass is produced mainly in medium (48%) and very big (29%) production systems, with medium (38%) or high and very high (together, 33%) production intensities. The pathway followed by most enterprises (57%) is the substitution of fossil resources, meaning mainly bioenergy production.

They are generally larger companies: 33% of them have more than 100, and 24% more than 500 employees. The technological levels are low to medium, as they use generic, internationally recognized technology, which is why these activities can be replicated in different places without any inconvenience. The companies of this cluster maintain a high level of cooperation with scientific and technological organizations from which they obtain information or with which they build their innovation processes, but more than anything else they maintain a high level of links with other companies and with local stakeholders, especially municipalities and provincial governments with which they maintain cooperation initiatives, especially for the creation and maintenance of infrastructures or for bureaucratic and administrative management.

These companies mainly produce for the national (47%) or, especially in the case of the larger ones, international markets (29%). Meanwhile, their main suppliers are national (76%). The reason for this could be that in the biomass-related sector, the requirement for highly sophisticated equipment is not that high, or that Argentine suppliers have already caught up technologically and specialized their production, so that they are able to offer the required modern technologies. Most companies (62%) stated that they attach great importance to the preservation of the environment and the care of natural resources

(81%), perhaps because they depend on local biomass production. In this sense, these are companies that have a high level of embeddedness, as also suggested by the most often mentioned key words "local development" (18%), "sustainability" (15%), and "added value" (15%). This level of anchoring in the territory is also manifested in the location criteria proposed by the actors: nearly all (95%) of enterprises, or 43% of total answers, mentioned as one of the main location criteria the availability of raw materials. This is one of the reasons why most of these companies are located in the Pampas (48%) and the NOA (24%) region.

Correlation analysis of the variables, using Spearman's Rho, allows for the identification of certain key elements within this bioeconomic approach. Firstly, there is a strong relationship between the volume of biomass used and the size of the companies. (0.523*), meaning that enterprises with higher biomass volumes tend to be larger. This is not really surprising, but it is a peculiarity of the biomass cluster, not relevant for the other clusters, where size is not significantly correlated with biomass use. Secondly, a strong relationship ($-0.477$*) can also be observed between size and cooperation with the private sector (Technology 5), meaning that enterprises with bigger size tend to have more private sector cooperation. Again, this result is not unexpected, but it differentiates the biomass group from the other two clusters, where no strong and only insignificant correlations exist. Thirdly, the correlation index also allows us to observe that there is a strong correlation between company size and cooperation with local stakeholders ($-0.478$*), i.e., the larger the companies are, the more links they have with these stakeholders. This correlation is also negative for Cluster 2 and 3, but less strong ($-0.378$ and $-0.162$, respectively) and not significant ($p = 0.165$ and $p = 0.635$). Fourth, from a technological point of view, significant correlations can also be observed between Technology 1 and Technology 4 (0.433*), and 5 (0.483*), meaning that the more biotechnologies are used, the more scientific and private sector cooperation exist. This is quite logical as these companies require the support of science and technology centers to drive high-tech processes. These correlation are not strong and insignificant for Cluster 2, the biotech group, probably because the values for Technology 1 are all very high (level 4 or 5) for this cluster, and because these companies have their own R&D teams, with less need to rely on other partners. The same holds true for the correlations between Technology 1 and Territorial 4 ($-0.584$**) or Territorial 5 ($-0.507$*), meaning that use of more biotechnologies is associated with more local identity and more environmental preservation. Finally, the biomass cluster has strong and significant correlations between Technology 5 and Territorial 3 ($-0.527$*), Territorial 4 ($-0.438$*), Territorial 5 ($-0.668$**), and Territorial 6 ($-0.438$*), implying that more private sector cooperation goes along with less influence of international prices, more local identity, more environmental preservation, and sustainable use of natural resources. In sum, it seems that within the biomass-focused group two tendencies could be detected: as size increases, cooperation with the private sector (for example, with local biomass producers) and with local stakeholders (such as local authorities) becomes more important, and this translates into more local embeddedness. Additionally, the more biotechnologies are used, the more important cooperation with other sectors becomes, for example with local R&D centres, which also leads to stronger local embeddedness.

Because of these strong correlations, an intra-group cluster analysis was carried out, using Size, Technology 1, and Biomass 1 as variables, which allowed the observation that there are three sub-groups within this bioeconomic approach: one composed of large companies with a low technological level, especially linked to the basic production of biofuels; another of medium-sized companies with high technology levels, linked to productive integration with various products; and a third subgroup of small companies with low technology, mainly oriented towards the production of food and some type of basic processing. An example of the first group is a large company which produces sugar and its by-products, especially alcohol and bioethanol, and which exploits the demand for biofuels, using tested technology. An example of the second group is a fully integrated company which produces cereals, oilseeds, and meat intensively, and produces energy

and waste from maize production, which is used for animal feed, and the waste from animal production is then used to generate electricity. An example of the third group is a company which produces juice concentrates and essential oils on a large scale, but also with a technology that is already well known on the market, using the large volumes of fruit and fruit waste production in their area.

### 4.2. Cluster 2. Biotechnological Approach

The companies that make up this cluster are characterized by their emphasis on the generation and application of modern biotechnological knowledge, supported also by a strong relationship with scientific and technological organizations present in the country (INTA, CONICET, INTI, among others) and by having international patents. The pathway most companies follow is the low volume–high value (40%) or the productivity-enhancement (40%) pathway. Knowledge is key to this cluster, so the keywords that characterize these companies are innovation, R&D, and biotechnology. In this bioeconomic approach, no (20%) or only few (73% of companies) biomass is used, which is mainly locally sourced (53%) but can also come from different parts of the country. The companies are mainly oriented towards the generation of specialized (43%) and niche (48%) products to improve the productivity of the agricultural sector in Argentina (seeds, liquid fertilizers, biostimulants, etc.), and pharmaceutical products. They are (very) small (66%) or, on the contrary, medium (27%) and very large (7%) companies, depending on their level of development. They are closely linked to national and international markets, both for the purchase of inputs and for the sale of their products.

The importance of innovation, R&D, and the relationship with universities or scientific and technological centers is also reflected in the importance of certain localization factors that these companies consider, such as local connections (24% of total answers) and the availability of skilled labor in the territory (14%). The need for good scientific and technological networks has determined that these companies are located especially in the Pampa region and in the metropolitan area of Buenos Aires.

There are some strong and important correlations in this bioeconomic approach, such as between Size and Territorial 4 ($-0.610^*$) and Territorial 6 ($-0.564^*$), meaning that bigger companies tend to have more local identity and more sustainable use of natural resources. Also, there was a strong correlation between Territorial 6 and Territorial 7 ($0.705^{**}$), meaning that more sustainable use of resources goes along with more cooperation with local stakeholders, or vice versa. In sum, the tendency in Cluster 2 seems to be that bigger companies are more locally embedded.

This led us to do another cluster analysis using the above cited variables with significant correlations, and detecting two sub-groups of companies: one of smaller companies that have a lower level of embeddedness, and another group of larger companies that have a higher level of embeddedness, which is a finding in contrast to the existing assumptions that the larger companies are alien or not linked to the rural territories. An example of the first group is a company which develops and manufactures formulations for pharmaceutical, dermo-cosmetic and nutraceutical industries aiming for the higher quality standards that each industry requires, and another one which develops inoculants, bio-controllers, and growth promoters for the agricultural sector. An example of the second group is a company which develops and commercializes in Argentina and in the international market in vitro diagnostic reagents for human health, biological research, and agro-biotechnology (animal health, plants, and seeds).

### 4.3. Cluster 3. Bioembedded Approach

The third bioeconomic cluster is clearly different from the biomass and the biotechnology approach. The approach referred to as the locally embedded bioeconomy approach, in short, the bioembedded approach, is characterized by using small amounts of biomass (82%) but of local origin (82%), having low levels of biotechnology use, and being (very) small sized (73%). They are mostly involved in the production of value-added food and

bioproducts that make use of local or national biomass, and also make use of or recycle waste from other activities. These are generally niche (45%) or specialized (36%) products, with very clearly identified markets. Two elements that are very important for this approach are the attention to the protection of natural resources and the importance they attach to the local identity of their products or processes, which is why designations of origin or local quality seals are often used. The construction of a new model of local productive development, more respectful of the environment and the circular economy, is also evident in the key words most frequently repeated by these actors, where value addition, local development, and sustainability appear as the main business objectives. This concern for the environment, identity, and the construction of local development processes is also manifested through the criteria for the location of these initiatives, where the social and environmental quality of the place (25%) comes first, followed by the availability of connections with the territory (21%), and only thirdly by the supply of raw materials (17%). Cluster 3 is also distinguished by the main pathway it follows, P5, which is the innovation of products and services that create local value added. These initiatives are distributed throughout the country, although there is a greater concentration in Patagonia, a region that is seen as a fertile territory for the generation of new bioeconomic initiatives, linked to nature and organic production.

However, despite these basic characteristics, certain differences can be found within this group. The correlation analysis carried out on the variables of this cluster allows us to observe that in technological terms there is a strong correlation evident: the companies with more patents have more scientific cooperation (Technology 3 and Technology 4, 0.623*). This allows us to observe that there are two types of companies within this cluster, those that have lower technological levels, such as natural foods and/or their derivatives, and those that operate with higher technological levels, especially those that produce biomaterials. Companies of the first group comprise, for example, a company producing caiman skin and meat in a "ranching system", in partnership with local communities. The high-value skins are exported to different markets, and the meat is consumed in the domestic market. Other companies produce special, totally natural sauces and dressings, and organic food, e.g., high quality hazelnuts. The second group contain companies producing bioplastics from sugar, cellulose and proteins, and a company producing packaging, bags, and other compostable, organic products.

## 5. Discussion

(1)  Argentina's bioeconomy is path dependent, with a predominance of the biomass–biotech approaches, but new development paths with more socio-ecological traits are opening up.

In relation to the main purpose of this paper, which is to identify the diversity of bioeconomic approaches in Argentina, and make alternative models more visible for policy debate, the analysis has shown that the bioeconomic clusters identified are consistent with the history and production model of Argentina, a country endowed with large biomass resources, an agricultural tradition, and the presence of solid scientific and technological networks [32]. This is manifested in the presence of Clusters 1 and 2. In short, these two clusters mark two key elements, firstly that there is a strong availability of biomass that is beginning to be exploited in a much more comprehensive way by multiple activities through mechanisms for generating added value (production of oils and bioenergy), not only substituting products, but also generating new products and processes [28]. Secondly, there is a strong demand for products to boost agricultural production, in order to make Argentine agriculture much more competitive, which is clearly visible through Cluster 2 innovations of new processes for higher agricultural productivity. Both clusters 1 and 2 have different levels of complexity, one takes direct advantage of biomass resources, the other builds on agriculture but enhances it through knowledge-based innovation of new products and processes [28]. Apart from their differences, it is clear that both clusters

indicate a path dependency [33] where synergies between the two lead to increased primary sector productivity, driven by innovations in the primary sector and in the biotech industry.

Cluster 3 emerges, on the one hand, as a product of a new look at natural resources and the environment. The development and growth of this sector of the bioeconomy is directly related to the new demand for organic products, new forms of consumption, and new forms of production that reduce waste and the environmental impact of production processes [34]. However, on the other hand, the emergence of this cluster is an important phenomenon as it indicates that the bioeconomy in Argentina does not necessarily only rely on traditional production sectors linked to the agricultural sector, or on the availability of biomass, but is opening up to new bioproducts or non-traditional or niche foods [5], which means that these activities are often not only located in rural areas, but increasingly in cities of different sizes, closer to consumer markets, or in areas with a stronger environmental protection and which cares about its identity, like the Patagonia region. Although this cluster still has lower levels of development than the other clusters built on the strong availability of biomass and the agricultural tradition, we expect that this bioeconomic approach will expand in the coming years due to the growing demand for bio-products and more specialized or niche foods. Yet, this might face challenges of behavioral innovations described by Bröring et al. [28].

However, there is one key element that has enabled the development of the bioeconomy in Argentina, namely the availability of universities and R&D, which has enabled the development of multiple bioeconomic activities [14]. Although the research did not focus on assessing the importance of Argentina's scientific and technological apparatus, a large part of the companies surveyed maintain different types of links with the scientific and technological system. This is most evident in Cluster 2, which depends on the generation and dissemination of innovative and modern knowledge, but surprisingly also in Cluster 1, which reflects a positive trend in the innovation capacity of traditional biomass-based companies. Even if Cluster 3 has lower values in scientific cooperation than Cluster 1 and 2, more than half of Cluster 3 enterprises use patents. However, despite the differences in the level of knowledge and technologies used, and despite the different scientific cooperation strategies of each cluster, it is important to note that most ventures of all clusters are highly prone and oriented towards the use of new knowledge as a key factor in the bioeconomy [3].

(2) Bioeconomic approaches in Argentina are partly consistent with contemporary conceptual approaches, but there is diversity within the clusters which makes a more differentiated analysis of the approaches worthwhile.

Answering our first and second research question, whether different bioeconomic approaches can be clearly distinguished and what their characteristics are, we found certain concordance between the theoretical models described above and the clusters identified in Argentina. Following Bugge et al. [5], there is a focus on the use and upgrading of biomass in Cluster 1, on biotechnologies and the use of research & technology in Cluster 2, and on high-quality products with territorial identity in Cluster 3. This cluster is also characterized, as the literature points out, by having small-scale production units, shorter supply chains, and strong relations with local stakeholders [30]. The identification of these three clusters were possible by using only two key variables: biomass volumes, and level of biotechnology used. A cluster analysis including all of the 14 variables did not reveal any clear patterns, indicating that in our sample, there is not such a clear continuum from a biotechnological to a socio-ecological way, as depicted in Table 1.

This fact can also be detected in Table 4 (and in Figure A1 of Appendix A), where two things stand out and deserve attention: first, that there is a strong intra-cluster diversity for some variables, and second, that there does not seem to be much difference between the clusters for some variables. For example, intra-cluster analysis showed that in Cluster 1 there are sub-groups with different company size and technological levels, and that this translates into more or less local embeddedness, i.e., more or less cooperation with local business, local R&D centers, or other local stakeholders. For Cluster 2, there seem to be a group of smaller companies with a lower level of embeddedness, and another group of

larger companies with a higher level of embeddedness. Finally, also in Cluster 3, there seem to be at least two types of companies, depending on the technological level. One explanation for the intra-cluster diversity could be what Mac Clay and Sellare [27] described as different stages of value chain upgrading, with different levels of biomass use, technological innovation, investments, risks, cooperation, and knowledge-sharing. Some companies of each cluster might be more advanced in the upgrading process than others, leading to different characteristics and also, to different economic and environmental outcomes.

Another reason for the intra-group clustering might be the sectorial affiliation of the enterprises, as described above. It seems that the sector characteristics overlap with the characteristics of the clusters. The sectorial dimension of bioeconomy models is not explicitly discussed by the authors presented in the theoretical background section [5–7,16]; only Müller and Korsgaard [31] mention in their typology for embeddedness some prevalent sectors such as tourism or specialty food and beverages. Our research shows that in Argentina, there seem to be significant differences in terms of sectors dominating the three distinguished clusters: the bioenergy sector prevails in the biomass, the agro-input sector in the biotech, and the food and the biomaterial sectors in the bioembedded approach. The other, less predominant sectors within the clusters might then have different characteristics, be it size, technological level, or embeddedness.

The intra-group diversity also means that for some of the variables, differences between the groups are not very pronounced. For example, no significant differences between the three clusters could be found in the origin of biomass, in the importance of local knowledge, in the use of patents, or in terms of target markets and the influence of international prices. Expressed in another way, this means that in Cluster 3, too, highly technological, not just local and traditional, knowledge is used [5–7,16,23], and that its products also serve and are dependent on the national or international markets, and that Cluster 1 and 2 do not always consist of multinational companies acting on global value chains [5,6,16,23] or are based on the commodification of knowledge and use of patents [16,23], contrasting with some of the bioeconomic typologies described in Table 1. This calls for a more differentiated discussion on the characteristics of the approaches. For example, it would be highly interesting to have a closer look on small-sized enterprises of Cluster 1, or on enterprises that depend on local, traditional knowledge of Cluster 2, or on large sized enterprises of Cluster 3 which use patents, etc.

(3)    All bioeconomic approaches are strongly linked to the territory where the respective companies are present, but the clusters are locally embedded in different ways with implications on possible sustainability outcomes.

Regarding the third research question, what kind of links the different approaches maintain with rural territories, and what sustainability impacts they have, it was possible to observe different relationships established between bioeconomic types and rural areas [35]. All bioeconomic clusters exert local linkages and play a role in local development, albeit to a different extent and for different reasons. This raises the question if local embeddedness strictu sensu can be claimed by all clusters, or if it is, as we argue, a key characteristic of the third cluster.

The first cluster has local linkages through the high volumes of biomass used. The difficulty of mobilizing large volumes of biomass encourages local and regional production and therefore the development of supply chains in the same territory [36], even if it is not clear to which maximum distance biomass can be transported without the costs inhibiting the profitability and development of the enterprises. It can be supposed that the development of the bioeconomic activities of this cluster promote the construction of very dense productive networks, generating new jobs and boosting local development, but also the enrichment of the local productive fabric allows the improvement of the socio-economic conditions of the territory, improving the attractiveness of the territory, and the anchoring of income at local level. In this sense, this cluster can be a generator of virtuous cycles of development in its own territories, provided that the conditions for its development are met, such as, in addition to the presence of biomass resources, the availability of certain

basic infrastructures, and the availability of qualified human resources for these activities. However, this model is often linked in Argentina to monoculture production systems, which can lead to land use change, biodiversity loss, and other adverse socio-ecological consequences. However, some of the companies also use organic side streams or waste as inputs, making a more efficient and cascading use of locally available biomass.

The second cluster is, as Korsgaard et al. [22] point out, an active participant in the globalized flow of resources, services and products across multiple locations given that it buys inputs or sells its products in multiple locations, but it also keeps links to the territories, partly because rural territories are the places where biomass is supplied, and also where agro-inputs (seeds, fertilizers) are needed. This cluster is also linked to the territory because the companies build strong relationships with the scientific and technological research centers and networks operating in the nearest cities, as in the model of the Italian industrial districts. These clusters contribute to build a greater exchange and mobility between the countryside and the city, which has allowed numerous small and medium-sized cities in Argentina to grow significantly in the last decades, often driven by the location of these types of companies and other services linked to them [37]. However, this model is often less biomass demanding, and therefore, on the one hand, creates less linkages to local biomass producers, but, on the other hand, may have less impacts on the environment. Moreover, it is often linked to an increase in agricultural productivity, and might lead to reduced land requirements.

The third cluster has a strong relationship with the territory, as it involves smaller companies that require biomass and local inputs, but above all because these companies need strong links with other private actors and with local and provincial governments, which support them through different mechanisms to be viable, i.e., their competitiveness depends on the networks they can build with other local actors to sustain themselves. This cluster also maintains a strong relationship with the territory due to the fact that many of the products generated include quality seals or have a local identity recognition, which expresses the close relationship between products and territory [38]. Following the reflections of Müller and Korsgaard [31], one can affirm that Cluster 3 initiatives have a special capacity to articulate global dynamics (markets, cultural, and consumer trends) with the capacities and characteristics of the territory ("bridging"), due to the fact that many of the products are either linked to international markets, or require global scientific knowledge, or require specific inputs imported from other countries. This articulation can often operate as a key factor in unlocking new opportunities for the development of the territory itself. Moreover, the ventures of this approach often explicitly follow agro-ecological principles, want to preserve the environment, and use only natural or recycled ingredients.

Considering these characteristics, we refer to Cluster 3 as the locally embedded bioeconomic approach, or the bioembedded approach, as it comes closest to the features described in the literature. This is not to say that the other approaches do not have linkages and are somehow locally embedded. On the contrary, despite existing assumptions that bioeconomic activities, especially those most dependent on biotechnology, are activities that tend to operate without relations to the local territory, and do not to generate dynamics of local development, the research shows that they build links that are key to territorial development. However, the three approaches contribute in different ways to the embeddedness and the development of the territories. Even if the analysis of the cases does not allow us to observe differences in the forms of action of the different entrepreneurs in relation to rural territories, which distinguish, as Korsgaard et al. [22] point out, "entrepreneurship in the rural" from "rural entrepreneurship", and further detailed research on the behavior of entrepreneurs is needed to investigate the different territorial logics of entrepreneurs, it can be postulated that the bioembedded approach, based on rather small-scale units, and with a strong local identity, follows a pathway of generating rather low-tech innovations that utilize resources locally available, thereby adding value, bridging local products to non-local customers, and contributing to a more circular, sustainable economy.

## 6. Concluding Remarks

The bioeconomy in Argentina is still strongly following the biomass and biotechnology approach, a path dependency which has developed over the last three or four decades, and which will certainly continue in the years to come. However, the bioeconomy is becoming more diverse. Different bioeconomic approaches will further develop, and probably coexist, in the short and medium term. Although the clusters identified show clear differences in the use of biomass, in technology, in the size of the companies, among other variables, there are two common elements in all clusters. Firstly, the interest in sustainability, the protection of natural resources, and innovation as a path to development, and secondly, the need of building networks and synergies with other actors or companies in the territory to generate better conditions for their own development and competitiveness. This means that there is a clear will to create greater embeddedness, which is considered a basic pillar of sustainability. These two observations suggest that the different bioeconomic initiatives could be setting the course towards a new model for the development of rural territories in Argentina, given that the model in force in recent decades has been characterized by a logic of little cooperation and articulation between economic agents, very little attention to the sustainability of natural resources, and especially very little concern for the future development of the territories. In fact, the bioeconomy appears as a new opportunity for territorial development in Argentina [38]. This will also require developing and implementing, besides general policies to foster the bioeconomy in Argentina [15], specific programs tailored to the needs of the different bioeconomic types. A good example of how a circular bioeconomy could be strengthened at the local level is the production of biomethane. The installation of biomethane plants requires the involvement and dialogue with local stakeholders, and offers the opportunity to achieve territorial energy self-sufficiency through small-scale systems [39].

However, we have to admit that our sample of 47 cases was rather small, and based on lists that were probably not systematically elaborated. For example, there were very few purely agro-ecological ventures in our sample, a fact that might have biased the characteristics of Cluster 3. We used an online survey with predefined answers, where it was not possible to ask further questions for clarifications or to go into detail. The survey used was opinion-based, so it reflects the subjective estimations of the interviewees on their local embeddedness, etc.

Having said that, we outline further research needs arising from this paper. A new research focused on the forms of linkage between bioeconomy and territory would allow to identify and analyze the type of relationship established between the development of the bioeconomy and the development of rural territories, as this will allow to argue for the need to deepen policies for territorial development through bioeconomic development strategies. In this sense, we put forward a hypothesis for rural development in Argentina: bioeconomic activities can be a clear factor in the development of rural territories, as they have a strong capacity for embedding, and to create virtuous cycles of development in rural areas, superior to the traditional extensive agricultural activity in Argentina, not only because of its capacity to use local resources, but also because of its capacity to link local dynamics with global markets while, at the same time, creating links to local stakeholders and a sense of local identity. This could be especially true for the bioembedded approach. Even if this approach currently might have limited relevance for territorial development in Argentina, it can make a significant contribution in the future and become an opportunity for a country deeply marked by territorial imbalances. Stronger support from public institutions is needed to promote this approach. However, such public support might not be enough. A solid social and productive base in the territories seems to be indispensable for the scaling up of those initiatives [40]. To explore these hypotheses further, it will be important to look more closely at the business models, local value addition, and territorial socio-economic and ecological impacts of the ventures of the different approaches. Our research could only give some qualitative indications of these developments, and more quantitative data are needed. In addition, research is needed on the social impacts of the

different bioeconomic approaches, e.g., in relation to aspects such as food security, health, and gender. Future research should also look more closely at agro-ecological initiatives, so that the picture of the Argentine bioeconomy becomes even more diverse, and appropriate support policies can be tailored to the needs of these ventures.

**Author Contributions:** Conceptualization, J.D. and M.S.; methodology, J.D. and M.S.; software, J.D.; validation, J.D. and M.S.; formal analysis, J.D. and M.S.; investigation, J.D. and M.S.; resources, J.D. and M.S.; data curation, J.D. and M.S.; writing—original draft preparation, J.D. and M.S.; writing—review and editing, J.D.; visualization, M.S.; supervision, J.D.; project administration, J.D. and M.S.; funding acquisition, J.D. All authors have read and agreed to the published version of the manuscript.

**Funding:** This research has been funded by the German Federal Ministry of Education and Research within the project STRIVE (Sustainable Trade and Innovation Transfer in the Bioeconomy, grant number BMBF: 031B0019, www.strivebioecon.de, accessed on 9 September 2022), and of the Bioeconomy Science Center of the Federal State of North Rhine Westfalia within the Transform2Bio project (Integrated Transformation Processes and their Regional Implementations: Structural Change of Fossil Economy to Bioeconomy, grant number BioSC: 53F-50000-00-13050200, https://www.biosc.de/transform2bio, accessed on 9 September 2022). This work was supported by the Open Access Publication Fund of the University of Bonn.

**Institutional Review Board Statement:** Not applicable.

**Informed Consent Statement:** Informed consent was obtained from all subjects involved in the study.

**Data Availability Statement:** Data can be made available upon request.

**Acknowledgments:** The authors gratefully acknowledge the contributions of team coordinator Jan Börner, and the statistical advice of Andrés Meiller. The authors further thank all the interviewees of the enterprises in Argentina who dedicated their time and gave us valuable information. We are also grateful for the valuable comments from Jorge Sellare, Pablo Mac Clay, and from the anonymous reviewers.

**Conflicts of Interest:** The authors declare no conflict of interest.

# Appendix A

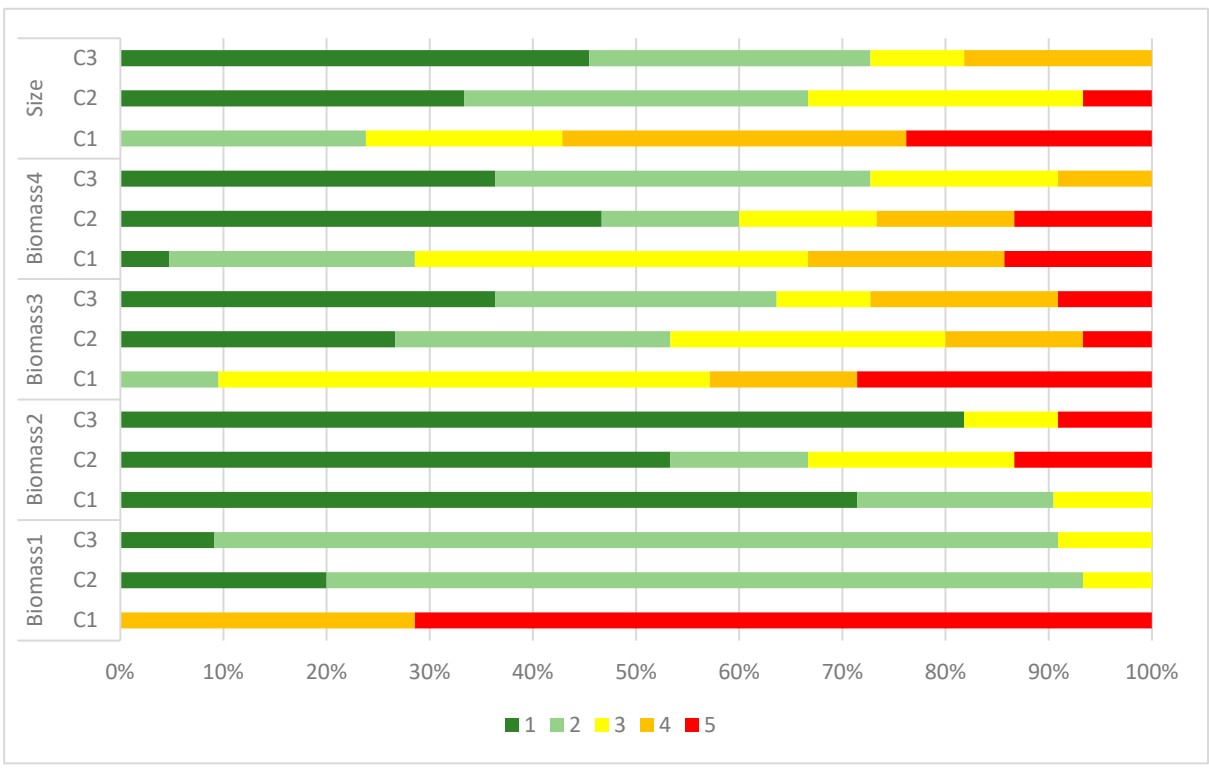

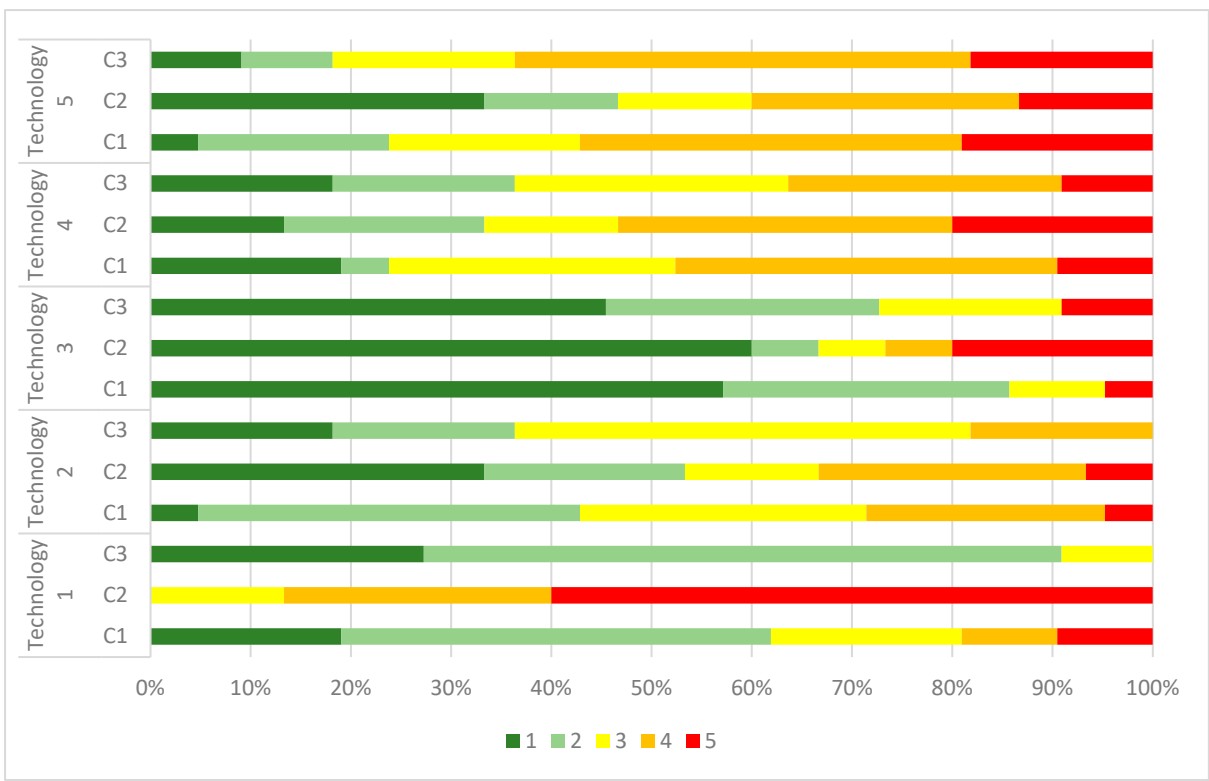

**Figure A1.** *Cont.*

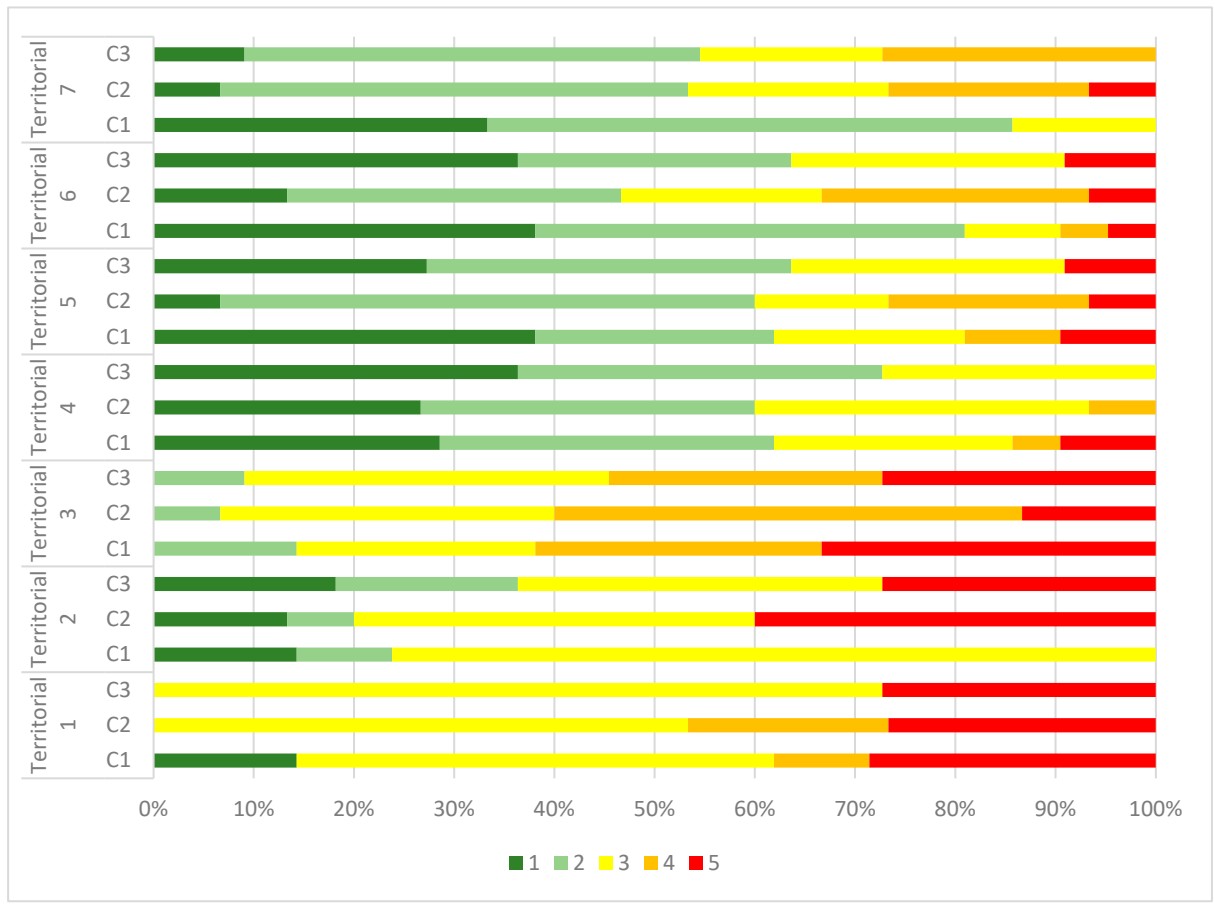

**Figure A1.** Distribution of enterprises belonging to 5-point scale, per variable and Cluster C1, C2 and C3.

**Appendix B. Online Interview Questionnaire (Translation from Spanish)**

Enterprise:

Location:

Province:

Web page:

**Sector**

1. Which products and/or services do you generate in your company? (ordered by importance, free text)
2. Which type of products or services do you produce:
   - Products that substitute fossil fuels
   - Products that enhance the productivity of the primary sector
   - Products that allow for a better and more efficient use of biomass
   - Products of low volume and high value
   - New products with local value added
3. Please, select FOUR key words of the list which best characterize your venture:
   - Local Development
   - Biomass
   - Biotechnology
   - Circular economy
   - Biodiversity
   - Investigation and development
   - Soil fertility

- Competitiveness
- Sustainability
- Patents
- Value addition
- Innovation

4. How are the products that you generate?

- They are generic products and there are many other producers on the market
- They are specialized products with little competition in the market
- They are niche products, very specific, there are no other companies that generate them at a national level

**Biomass**

1. What total amount of biomass (in tons) do you use per year?

| Zero | Less than 10 t | Between 10 and 100 t | Between 100 and 1000 t | More than 1000 t |
|---|---|---|---|---|

2. Where does most of the biomass you use come from?

| Local | Province | National | Latin America | International |
|---|---|---|---|---|

3. What is the size of the agricultural/forestry/marine farms where the biomass that you use in your activity comes from?

| Very big | Big | Medium | Small | Very small |
|---|---|---|---|---|

4. How intensive is the use of chemical inputs, modern varieties and sophisticated machinery for the production of the biomass that your company produces and/or buys?

| Very intensive | Intensive | Medium | Low intensive | Not used |
|---|---|---|---|---|

5. What category do your main clients belong to (in percentages)?

- Industrial enterprises %
- Food companies %
- Agricultural service companies %
- Agricultural/forestry/fisheries producers %
- Final consumers %
- Government/State Organizations %
- Biotech companies %
- Human health companies/Laboratories %
- Others: %

6. The products that you generate are mainly for sale and use at which level:

| Local | Province | National | Latin America | International |
|---|---|---|---|---|

**Size**

7. What is your annual turnover?

| Less than 50.000$ | Between 50.000 and 250.000$ | Between 250.000 and 1 millon $ | Between 1 y 10 millon $ | More than 10 millon $ |
|---|---|---|---|---|

8. How do you consider the production scale of your company compared to most other companies in the same field in Argentina?

| Much bigger | Bigger | Average | Smaller | Much smaller |
|---|---|---|---|---|

9. How many people work in your company?

| Between 1 and 5 persons | Between 6 and 20 persons | Between 21 and 100 persons | Between 101 and 500 persons | More than 500 persons |
|---|---|---|---|---|

**Innovation and Tecnology**

10. How much of the value of your production depends on innovations and processes based on modern and standardized biotechnology?

| 0% | 1–24% | 25–49% | 50–74% | 75–100% |
|---|---|---|---|---|

11. How much of your production value depends on innovations and processes based on local experience and tradition?

| 0% | 1–24% | 25–49% | 50–74% | 75–100% |
|---|---|---|---|---|

12. How much of the value of your production is based on patents?

| 0% | 1–24% | 25–49% | 50–74% | 75–100% |
|---|---|---|---|---|

13. How important has been the cooperation with state scientific and technological organizations for the development of your products? (Universities, INTA, INTI, CONICET, others)?

| Very important | Important | Regular | Less important | Not important |
|---|---|---|---|---|

14. How important has cooperation with companies or private groups been for the development of your products?

| Very important | Important | Regular | Less important | Not important |
|---|---|---|---|---|

**Territorial Conditions**

15. **Excluding** biomass, where are most of your company's suppliers of inputs, goods and services?

| Local | Province | National | Latin America | International |
|---|---|---|---|---|

16. What were the **THREE** main reasons why you chose the place where your company is located?
   - Availability of raw material nearby
   - Workforce cost
   - Access to international markets
   - Availability of public infrastructure
   - Personal connections to the area
   - Availability of skilled labor
   - Accessible and cheap land
   - Proximity to research and development centers
   - Proximity to Universities
   - Subsidies
   - Social, cultural and/or environmental quality of the place
   - Others:

17. To what extent do international market prices influence the profitability of your business?

| Very much | Much | Somehow | Not much | Not at all |
|---|---|---|---|---|

18. To what extent are your products based on a local brand, with local identity and cultural recognition?

| Very much | Much | Somehow | Not much | Not at all |
|---|---|---|---|---|

19. To what extent have your activities or products contributed to improving the environmental quality of the area?

| Very much | Much | Somehow | Not much | Not at all |
|---|---|---|---|---|

20. To what extent have your activities or products contributed to a more sustainable use of the area's natural resources?

| Very much | Much | Somehow | Not much | Not at all |
|---|---|---|---|---|

21. How much does your company interact with local organizations and local civil society?

| Very much | Much | Somehow | Not much | Not at all |
|---|---|---|---|---|

22. Could you identify three companies that operate in the same field as you in Argentina?

**Appendix C. Map of Enterprises of the Different Clusters in Argentina**

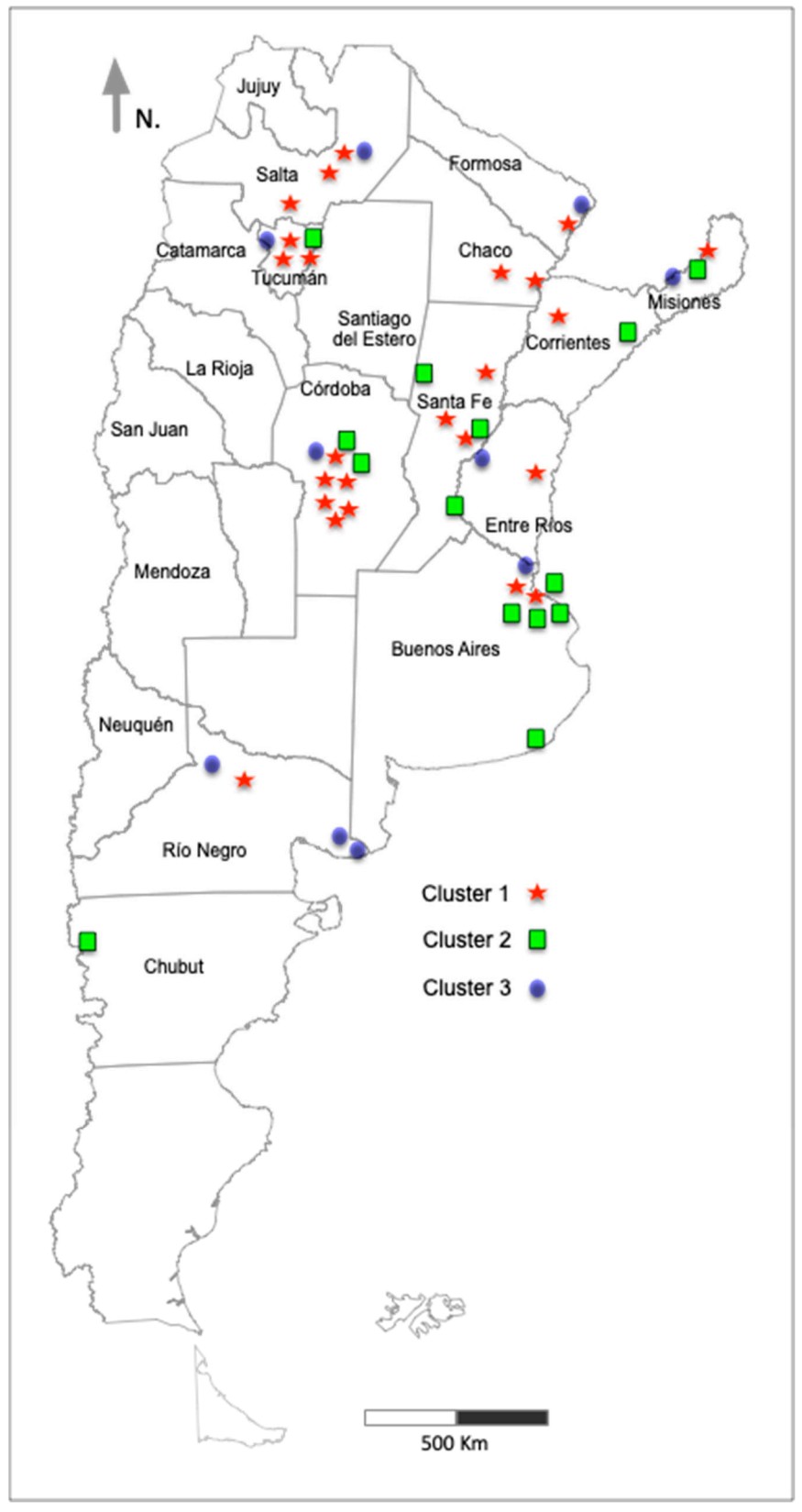

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
