# Peer review of "New or Traditional Approaches in Argentina’s Bioeconomy? Biomass and Biotechnology Use, Local Embeddedness, and Sustainability Outcomes of Bioeconomic Ventures"

_sustainability, doi:10.3390/su142114491_

Round 1

Reviewer 1 Report

Dear Authors, 

Congratulations on doing great research. I read it carefully and I did not find major concerns I recommend it for publication. 

I noted just one comment and please find it in the attachment. 

All the best,

Sajad

Author Response

Thank you very much for your careful revision and comments! We made the first Table clearer (what is meant by "Size"). We considered showing the correlation analysis results in a table, but it would be a real big table (19x19) with many non-significant results, so that we would like to describe only the significant ones in the text.

Reviewer 2 Report

Comments to authors:

The paper discusses different bioeconomy types in Argentina. It makes an important contribution to the development of more local bioeconomy initiatives and their local embeddedness. Something that would make an interesting contribution also in other geographical contexts and whether these initiatives are part of a more overall discourse on bioeconomy or if it is mere greenwashing of larger industrial sectors. Thus, I find this a very interesting paper that is well written and for being a first draft it is also well structured, even though there are some minor language editing that must be done. However, I also have some major problems with the paper that have to be dealt with before publishing.

Section 1.

I have a problem with your approach on bioeconomy typologies. At first you state that there are two bioeconomy development models (p. 2). In the next section (p.2 line 23) you claim that “This diversity of bioeconomic approaches have not yet been documented and analyzed in Argentina.”… This is a problem in general in your paper as you use throughout use concepts such as bioeconomy models, approaches, typologies, categories and types. This is a bit confusing for the reader. What is the difference between e.g. models and typologies or types and approaches? You use all of these concepts throughout your text and sometimes mention them together. Is there a difference or do you refer to the same thing? This is important in relation to the use of typology as a framework. Also, what do you mean when saying that there have been no attempts to classify bioeconomic ventures into bioeconomy typologies? What is the difference...?

On p. 2 line 35 you refer to scales. It is unclear what you mean by scale in this context.

By connecting to the comment on the contribution to your paper of using typologies this also connects to the papers RQ:s:

“Firstly, can the ideal types of approaches be clearly distinguished in the case of Argentina? Second, what are the characteristics of the bioeconomic approaches that could be identified? Third, which kind of links do these approaches maintain with rural territories, and what sustainability impacts derive from these linkages?”

In the first RQ, what is meant by ideal types of approaches? Ideal types as decided by whom and ideal types of bioeconomies or developing bioeconomies?

In all RQ:s you are talking about types of approaches, how this relate to the bioeconomy typology? Maybe a key could be to limit your text to the use of approaches and therefore go through your paper to be consequent and cut out models, types and categories or use one of the other ones.

Can you answer the second part of your third RQ by the survey you made? How do you draw conclusions on what sustainability impacts that can be derived from the linkages to rural territories?

I understand why you talk a about the local embeddedness of enterprises, but when you first present the organisation of your paper (p. 3, line 14-15) you start by saying that first you will describe the main bioeconomy typologies discussed in the literature, and the local embeddedness of enterprises. Here it is very unclear how and why these are presented like this in your “theory” section. Why is there first a typology introduced and why do you have a section on the local embeddedness of enterprises? Explain and argue why.

Section 2

It is not clear why you are using typologies or how. This must be better explained for the reader. How does the use of typology help your analysis and is it a tool for your analysis. You have to argue why you are not just talking about models, types, approaches that you mention earlier and that you anyway refer to throughout your paper. How does these models that you mention earlier turn into bioeconomy typologies, what is the aim of using typologies?

See e.g. Dobson (1995) Environment sustainabilities: An analysis and a typology.

Table 1. Here you present the different types of the bioeconomy, which is unclear if this is to be an overview of the typology of bioeconomies. It needs to be explained how you will use this table, and can you summarise it in anyway? You have also added your own types, which I see as part of your result. That should rather be added later or that one thing that you come up with from your study is another kind of typology than the one presented under section 2. This should rather be discussed in your results and your concluding remarks. As it stands now, it gets very confusing for the reader. However, I am a bit unsure how your own three bioeconomy models differ compared to e.g. Bugge et.al. Maybe this becomes clearer if you add it later in your text.

Section 2.2.

I think this is very important for the discussion on the development of bioeconomy and its local embeddedness. However, it is unclear why and how this relates to your earlier typology. Guide the reader on why and how the concepts are used. Also, all models to the right of your table seems to fit here. I think you have to try to identify and use the outcome of the table to discuss this. It gets a bit difficult to follow the text for the reader as you are going back to more specific models here.

Section 3 Methods

I miss a reflection on the possible problems with the way you have sampled the companies in your study. Earlier you mention that a large part of the bioeconomy is linked to GM and export orientation with a bio-technological and agroindustrial focus. How have you ensured that by using these lists you are also included the possibly marginalised firms within this sector?                                                                                                                 

On p. 7 line 6-7 you write “i.e. enterprises with characteristics mentioned in the previous section”. In what section? I can’t find these characteristics of the bioeconomy that you refer to or make it clearer to what you are referring.

Under section 3.2 Data collection, p. 7. make a reference to table 2.

Section 4. Results

On. p.10 line 9-13. This could be a bit further developed. You write that you named the clusters accordingly, but here you come back to model instead of type... You also write following those in literature, but you still name them differently, which is perfectly fine, but it must be made clearer.

A suggestion would be; Derived from the literature and the cluster analysis we have identified three bioeconomy types in our material: the biomass model, the biotechnological model and...

Section 5. Discussion

It is perfectly fine to organise this section in line with the RQ:s. But it is unclear why there is a 1, 2 and 3 here as it stands now. Are these answers to your research questions? Then you should explain that in the beginning of this section. It is a bit unclear what these headings 1, 2, 3 refer to. Even though you have a clear answer to the RQs, the reader should not have to go back to the RQs in the beginning of the text. Remind us or just write the RQs as under headings.

Author Response

Thank you for very much for your comments and valuable suggestions!

Section 1.

I have a problem with your approach on bioeconomy typologies. At first you state that there are two bioeconomy development models (p. 2). In the next section (p.2 line 23) you claim that “This diversity of bioeconomic approaches have not yet been documented and analyzed in Argentina.”… This is a problem in general in your paper as you use throughout use concepts such as bioeconomy models, approaches, typologies, categories and types. This is a bit confusing for the reader. What is the difference between e.g. models and typologies or types and approaches? You use all of these concepts throughout your text and sometimes mention them together. Is there a difference or do you refer to the same thing? This is important in relation to the use of typology as a framework.

Answer: You are right, this might be confusing. First of all, in literature itself, one finds different wordings such as “visions”, “approaches”, “narratives”, “ideal types”, “models”, etc. (see Section 2). Actually, we did not elaborate an own “typology”, so we now avoid the term now, but used typologies of other authors (and still use this term when referring to them).  

We used bioeconomic “approaches” and “models” interchangeably, and what we mean with these terms (which are also both used in literature), is mainly how the concept of the bioeconomy is commonly understood, i.e., how the bioeconomy is represented in the public discourse, described in strategies, and treated in policies. We now excluded the term “models” when referring to the three clusters we have detected, and only use “approaches” as a wider term. This might be more appropriate, as “models” connote that causal relationships of variables can be described for the different models, which is not the case. We still use “model” when referring to general discourses on bioeconomy.

Also, what do you mean when saying that there have been no attempts to classify bioeconomic ventures into bioeconomy typologies? What is the difference...?

What we mean when saying that there have been no attempts to classify bioeconomic ventures into bioeconomy typologies, is that no grouping of bioeconomic enterprises with different characteristics into the bioeconomic types has been done so far (what we did with our cluster analysis). We re-wrote the sentence to make it clearer.

On p. 2 line 35 you refer to scales. It is unclear what you mean by scale in this context.

Answer: size

By connecting to the comment on the contribution to your paper of using typologies this also connects to the papers RQ:s:

“Firstly, can the ideal types of approaches be clearly distinguished in the case of Argentina? Second, what are the characteristics of the bioeconomic approaches that could be identified? Third, which kind of links do these approaches maintain with rural territories, and what sustainability impacts derive from these linkages?”

In the first RQ, what is meant by ideal types of approaches? Ideal types as decided by whom and ideal types of bioeconomies or developing bioeconomies?

Answer: Ideal types in the sense of Weber, a term which some of the authors (Bugge, Vivien) of the literature review use. We now deleted this term in the RQ, as it is not appropriate here.

In all RQ:s you are talking about types of approaches, how this relate to the bioeconomy typology? Maybe a key could be to limit your text to the use of approaches and therefore go through your paper to be consequent and cut out models, types and categories or use one of the other ones.

Answer: You are right, the bioeconomic types we distinguish might not be enough to already talk of a “model”, and we also do not really develop a new typology. We now only talk of “types” or “approaches”, when speaking on “our” bioeconomic clusters, not any more on models or typologies. Only when citing literature, we use the terms the authors have used.

Can you answer the second part of your third RQ by the survey you made? How do you draw conclusions on what sustainability impacts that can be derived from the linkages to rural territories?

Answer: you are right, we cannot directly derive sustainability impacts from local linkages, but rather look at linkages and impacts together. (Two variables looked at sustainability impacts.) We reformulated the RQ.

I understand why you talk a about the local embeddedness of enterprises, but when you first present the organisation of your paper (p. 3, line 14-15) you start by saying that first you will describe the main bioeconomy typologies discussed in the literature, and the local embeddedness of enterprises. Here it is very unclear how and why these are presented like this in your “theory” section. Why is there first a typology introduced and why do you have a section on the local embeddedness of enterprises? Explain and argue why.

Answer: Thank you for the hint to make it clearer: the different bioeconomic approaches discussed in literature do not explicitly include the local embeddedness of the enterprises. We try to merge this concept with the bioeconomic approaches, something new. We now state this explicitly: “To our knowledge, the concept of local embeddedness has not been taken into account in the characterization of bioeconomic approaches so far, something we want to change in this article.” (p. 3) and explain it on page 7 (last paragraph).

Section 2

It is not clear why you are using typologies or how. This must be better explained for the reader. How does the use of typology help your analysis and is it a tool for your analysis. You have to argue why you are not just talking about models, types, approaches that you mention earlier and that you anyway refer to throughout your paper. How does these models that you mention earlier turn into bioeconomy typologies, what is the aim of using typologies?

See e.g. Dobson (1995) Environment sustainabilities: An analysis and a typology.

Table 1. Here you present the different types of the bioeconomy, which is unclear if this is to be an overview of the typology of bioeconomies. It needs to be explained how you will use this table, and can you summarise it in anyway?

Answer: Thank you for this comment, we now re-organized Table 1 to adjust it to our purpose, and explain how we used the typologies presented in Table 1 in an extra paragraph (p. 5).

You have also added your own types, which I see as part of your result. That should rather be added later or that one thing that you come up with from your study is another kind of typology than the one presented under section 2. This should rather be discussed in your results and your concluding remarks. As it stands now, it gets very confusing for the reader. However, I am a bit unsure how your own three bioeconomy models differ compared to e.g. Bugge et.al. Maybe this becomes clearer if you add it later in your text.

Answer: we now excluded our own model here and discuss in the Discussion Section in how far our clusters relate to the concepts of the authors (p. 17f).

Section 2.2.

I think this is very important for the discussion on the development of bioeconomy and its local embeddedness. However, it is unclear why and how this relates to your earlier typology. Guide the reader on why and how the concepts are used. Also, all models to the right of your table seems to fit here. I think you have to try to identify and use the outcome of the table to discuss this. It gets a bit difficult to follow the text for the reader as you are going back to more specific models here.

Answer: We now added a paragraph on this (page 7, last paragraph of section 2.2):

This means that enterprises with high local embeddedness might share many characteristics of the socio-ecological approaches (see Table 1), for example, minimizing the use of external inputs and showing local identity, mentioned by Bugge et al. [4] and Levidow et al. [21]. Also, the importance of regional value chains, autarchy, and the connection to local stakeholders [14] aims in this direction. But, by bridging local resources to non-local costumers, there might be also cases which rather belong to the approach of biomass transformation into marketable products [4] by replacing fossil resources by biomass [6], and cases where enterprises are active on international markets, a characteristic rather belonging to the bio-tech model [14]. We therefore included local embeddedness as an additional category for characterizing bioeconomic approaches.

Section 3 Methods

I miss a reflection on the possible problems with the way you have sampled the companies in your study. Earlier you mention that a large part of the bioeconomy is linked to GM and export orientation with a bio-technological and agroindustrial focus. How have you ensured that by using these lists you are also included the possibly marginalised firms within this sector?             

Answer: yes, there are problems with the sample, and we stated this explicitly: “nothing can be said about possible selection biases” (p. 7), and also in the limitations: “However, we have to admit that our sample of 47 cases was rather small, and based on lists which were not systematically elaborated. For example, there have been very few strictly agro-ecological ventures in our sample, a fact that might have biased the characteristics of Cluster 3.” (page 19) Nevertheless, the 47 respondents represent a wide array of enterprises (see Table 2).

On p. 7 line 6-7 you write “i.e. enterprises with characteristics mentioned in the previous section”. In what section? I can’t find these characteristics of the bioeconomy that you refer to or make it clearer to what you are referring.

Answer: we meant that there are enterprises which are biomass-focused, bio-tech enterprises, and also small-scale, agroecological enterprises, described in section 2.1. We tried to make it clearer.

Under section 3.2 Data collection, p. 7. make a reference to table 2.

Answer: sorry, but Table 2 does not belong to section 3.2, it is on sample structure!

Section 4. Results

On. p.10 line 9-13. This could be a bit further developed. You write that you named the clusters accordingly, but here you come back to model instead of type... You also write following those in literature, but you still name them differently, which is perfectly fine, but it must be made clearer.

A suggestion would be; Derived from the literature and the cluster analysis we have identified three bioeconomy types in our material: the biomass model, the biotechnological model and...

Answer: Thanks, we now do not use “model” any more, and rephrased as suggested.

Section 5. Discussion

It is perfectly fine to organise this section in line with the RQ:s. But it is unclear why there is a 1, 2 and 3 here as it stands now. Are these answers to your research questions? Then you should explain that in the beginning of this section. It is a bit unclear what these headings 1, 2, 3 refer to. Even though you have a clear answer to the RQs, the reader should not have to go back to the RQs in the beginning of the text. Remind us or just write the RQs as under headings.

Answer: Thanks, we now remind the reader at the beginning of each section as suggested.

Submission Date

19 September 2022

Date of this review

01 Oct 2022 08:17:57

Reviewer 3 Report

General comments

This is an interesting attempt to contribute to both the academic debate as well as policy rhetoric regarding understanding the diversity of bioeconomy ventures. One of the main strengths of this article is to demonstrate the importance of local embeddedness in the bioeconomy-related transition processes, particularly the discussion of how the concept of local embeddedness should be considered as a variable that can play a different role depending on the type of bioeconomy venture in question (biomass-based vs cooperation-based vs value-based). Although the boundaries between the different types of ventures understandably remain vague, an understanding of the importance of dense local linkages and a favourable institutional context for further development needs to be highlighted.

Title

The title is currently quite broad and gives somewhat misleading impression that the article is about the creation of a new typology (while in its current format, the article is rather testing the existing literature to better understand the Argentina’s bioeconomy). The title could be richer in substance, especially if the authors find a way to make the results more generalizable to other developing/transition countries.

Scientific relevance

The latter is reflected also in the framing of the research problem (see Introduction), which is very strongly oriented on practical policy-making problems Argentina is facing with (rather than the wider group of developing/transition countries that may face similar problems due to heavy reliance on traditional industries in the area of bioeconomy, etc). Therefore, the information provided for the case selection (Argentina) remains also limited (page 2, lines 29-33). The inclusion of in-text references from previous studies could be more systematic, including for framing the hypothesis (eg page 2, lines 24, 34).

Also in general, the greatest shortcoming of the article is related to the limited efforts to explain and build a bridge with the academic debate of the field: from opening a research puzzle in Introduction (page 3, lines 7-9) to building and explaining the analytical framework and its added value in the section 2.

Literature review and analytical framework

The section 2.1 remains surprisingly descriptive and lacks an assessment/critique of the different bioeconomy typologies that have been developed so far to provide a better understanding of the different nature of the bioeconomy. The discussion in Section 2.1 should provide a solid basis for developing the argumentation about the previous research in a comparative way. A stronger reliance on in-text referencing could also be expected. Also Table 1 needs further refinement for the assessment of the key focuses and characteristics of current bioeconomy typologies, as well as what is missing from the previous studies. E.g., it is worth mentioning that Hausknost et al.’s 2017 article also highlights the different key actors driving the different transition processes/scenarios related to the bioeconomy.

What is the basis for developing your own analytical framework – what are its main foundations and how do these pillars differ from those discussed in previous studies? Further, the analytical framework of this article should be clearly linked to the discussion in section 2.2 on local embeddedness and the dichotomy of “entrepreneurship in the rural” vs “rural entrepreneurship”.

The agro-ecological model is considered as the closest to the “ideal model”. It remains unclear, however, what is the basis for the normative preference for this model? Any criticism here?

Finally, while this articles attempts to unpack the nature of bieconomy ventures, it also remains questionable why the ground for operationalization relies on the chosen specific bioeconomy classifications. In the recent literature, we also find other types of typologies developed to understand important differences in the bioeconomy, eg Bröring, S., Laibach, N. and Wustmans, M., 2020. Innovation types in the bioeconomy. Journal of Cleaner Production, 266. https://doi.org/10.1016/j.jclepro.2020.121939

Methods

Given the effort that goes into explaining the intra-cluster differences, one might ask how much an overall response rate of 47% can actually affect the final results? Was it double checked which companies did not respond and were therefore excluded from the analysis?

Some clarifications are related with the Table 4 and the respective discussion:

First, is it fair to claim that companies in cluster 2 rely on strong scientific cooperation (pages 17-18), while according to the Table 4, the level of scientific cooperation in cluster 2 is lower than in cluster 1. What is the explanation here? On the other hand, it also reflects a positive trend in the innovation capacity of traditional biomass-based companies. Surprisingly, the scientific cooperation is the lowest in cluster 3.

Second, the reliance on national and international partners, as well as the actual level of dependence on technology/knowledge transfer from foreign countries, raised questions. Namely, it is claimed that cluster 1 enterprises rely upon national suppliers (excluding biomass) (page 10 line 25). Does this mean that the source of technological equipment etc is also national/local (rather than imported), considering here that this is one of the key means of process innovation in traditional industries.

The beginning of the section “Concluding remarks” needs further elaboration. The timeline in question needs to be specified, because reading the results of the survey, the impression of the current situation in terms of the cooperation intensity is not so negative?

Whether and to what extent the results/lessons learned are applicable in other developing / transition countries?

Policy implications

If one of the main aims is to contribute to the policy debate by demonstrating how bioeconomy ventures can differ due to the nature of economic activity and innovation strategies as well as the nature of local embeddedness, then one would also expect a discussion about more targeted policies regarding the different types of bioeconomy ventures (biomass and mass-production model, biotech-based model and bioembedded model).

In addition, the authors tend to overemphasize the relevance of bioembedded model (or agro-ecological model) in the transition processes, or at least, the role of these particular ventures is not sufficiently explained or supported by the data. E.g., see Table 4 and related explanations of how the innovativeness of this particular model is reflected merely via a focus on niche and specialized products? I.e., the reliance on biotechnologies or scientific cooperation does not seem to support the provision of innovative and premium level products in this cluster.

Specific comments

On page 9, lines 24-26, it is mentioned that four variables were decisive in the formation of three key models in the Argentine context, while on page 16, lines 28 and 29, it is claimed that these three models were formed based on only two variables?!

On page 7, lines 4-5, it is said that “It is not known how the lists exactly came about, …” – is this wording scientific enough?

Author Response

Thank you for very much for your comments and valuable suggestions!

Title

The title is currently quite broad and gives somewhat misleading impression that the article is about the creation of a new typology (while in its current format, the article is rather testing the existing literature to better understand the Argentina’s bioeconomy). The title could be richer in substance, especially if the authors find a way to make the results more generalizable to other developing/transition countries.

Answer: Thank you for your suggestion. You are right, it is not really a new typology, so we quit this in the title and choose another one. The second part of the title (Biomass, biotechnology and local embeddedness of bioeconomic ventures) might now be interpreted as a more general description of results.

Scientific relevance

The latter is reflected also in the framing of the research problem (see Introduction), which is very strongly oriented on practical policy-making problems Argentina is facing with (rather than the wider group of developing/transition countries that may face similar problems due to heavy reliance on traditional industries in the area of bioeconomy, etc). Therefore, the information provided for the case selection (Argentina) remains also limited (page 2, lines 29-33). The inclusion of in-text references from previous studies could be more systematic, including for framing the hypothesis (eg page 2, lines 24, 34).

Answer: We included more information on Argentina`s bioeconomy, and tried to be more systematic (p. 2). In relation to other countries, we are a bit reluctant to enter this discussion, as this would need much more information on other countries, which, for example for the embeddedness of bioeconomic ventures, is rather limited.

Also in general, the greatest shortcoming of the article is related to the limited efforts to explain and build a bridge with the academic debate of the field: from opening a research puzzle in Introduction (page 3, lines 7-9) to building and explaining the analytical framework and its added value in the section 2.

Answer: We tried to explain the framework and how it was developed in a better way (p. 5, last paragraph), see also below.

Literature review and analytical framework

The section 2.1 remains surprisingly descriptive and lacks an assessment/critique of the different bioeconomy typologies that have been developed so far to provide a better understanding of the different nature of the bioeconomy. The discussion in Section 2.1 should provide a solid basis for developing the argumentation about the previous research in a comparative way. A stronger reliance on in-text referencing could also be expected. Also Table 1 needs further refinement for the assessment of the key focuses and characteristics of current bioeconomy typologies, as well as what is missing from the previous studies. E.g., it is worth mentioning that Hausknost et al.’s 2017 article also highlights the different key actors driving the different transition processes/scenarios related to the bioeconomy.

Answer: we now made a comparative assessment of the typologies and made the Table 1 more systematic, giving key elements. Based on that, we argue why and how our approach was developed. We now mention that Hausknost et al. state that different actors (state, business, academia, civil society) not only have different visions, but also their roles for the transition to the bioeconomy have to be critically evaluated (p. 4).  

What is the basis for developing your own analytical framework – what are its main foundations and how do these pillars differ from those discussed in previous studies? Further, the analytical framework of this article should be clearly linked to the discussion in section 2.2 on local embeddedness and the dichotomy of “entrepreneurship in the rural” vs “rural entrepreneurship”.

Answer: We now explain better how our framework was developed (p. 5, last paragraph), and link it to the discussion on embeddedness (p. 7, last paragraph).

The agro-ecological model is considered as the closest to the “ideal model”. It remains unclear, however, what is the basis for the normative preference for this model? Any criticism here?

Answer: If this model is the ideal one, we do not propose, but we consider it necessary not to overlook this model. In the Conclusions, we make clear that this model could be an opportunity, as it is highly embedded locally, and needs more support, but not that the other models are less valuable. Further research is needed here.

Finally, while this articles attempts to unpack the nature of bioeconomy ventures, it also remains questionable why the ground for operationalization relies on the chosen specific bioeconomy classifications. In the recent literature, we also find other types of typologies developed to understand important differences in the bioeconomy, eg Bröring, S., Laibach, N. and Wustmans, M., 2020. Innovation types in the bioeconomy. Journal of Cleaner Production266. https://doi.org/10.1016/j.jclepro.2020.121939

Answer: Of course, the chosen classifications are specific, but we think they reflect well the current debate. We now also discuss the approach of Bröring et al., which is on innovation types of the bioeconomy (not on the bioeconomy as a concept) in the typology section (p. 6) and use it in our discussion (p. 16f.).

Methods

Given the effort that goes into explaining the intra-cluster differences, one might ask how much an overall response rate of 47% can actually affect the final results? Was it double checked which companies did not respond and were therefore excluded from the analysis?

Answer: We think that 47% is a very high response rate for an Online survey. The problem lies not so much in the response rate, but in the overall small size of the list (102 enterprises), which was explained and clearly stated as a limitation. Double check is not possible as we do not know the characteristics of the companies that did not respond.

Some clarifications are related with the Table 4 and the respective discussion:

First, is it fair to claim that companies in cluster 2 rely on strong scientific cooperation (pages 17-18), while according to the Table 4, the level of scientific cooperation in cluster 2 is lower than in cluster 1. What is the explanation here? On the other hand, it also reflects a positive trend in the innovation capacity of traditional biomass-based companies. Surprisingly, the scientific cooperation is the lowest in cluster 3.

Answer: Thanks for this good observation! We now included a possible explanation (p. 12, second paragraph):

“Scientific cooperation interestingly is higher for Cluster 1 than for Cluster 2, which could mean that biomass-related enterprises are nowadays searching for new knowledge and innovation capacities, whereas biotech companies might be more independent from public R&D institutions, as they often possess their own laboratories and research departments.”

And also in the Discussion (p. 18), following your suggestion: “This is most evident in Cluster 2, which depends on the generation and dissemination of innovative and modern knowledge, but surprisingly also in Cluster 1, which reflects a positive trend in the innovation capacity of traditional biomass-based companies. Even if Cluster 3 has lower values in scientific cooperation than Cluster 1 and 2, more than half of Cluster 3 enterprises use patents.”

Second, the reliance on national and international partners, as well as the actual level of dependence on technology/knowledge transfer from foreign countries, raised questions. Namely, it is claimed that cluster 1 enterprises rely upon national suppliers (excluding biomass) (page 10 line 25). Does this mean that the source of technological equipment etc is also national/local (rather than imported), considering here that this is one of the key means of process innovation in traditional industries.

Answer: This is a tricky question. First, yes, it means that the equipment is mainly from national suppliers, but it might be that the national suppliers import them, so that the origin would not be national, but we do not know. We added one sentence on this (p. 12 first paragraph).

Second, it would be interesting to examine this further, but we can only guess if it is a limitation for innovation processes, that suppliers are mainly national, or on the contrary, that Argentinean suppliers have already up scaled their technological level so that they are able to supply sophisticated technologies. We added one sentence on page 14, third paragraph.

The beginning of the section “Concluding remarks” needs further elaboration. The timeline in question needs to be specified, because reading the results of the survey, the impression of the current situation in terms of the cooperation intensity is not so negative?

Answer: Actually, it is hard to say if “the timeline in question” will be short, medium or long termed. The actual model developed over the last 30 years or so, and is still prevalent, see Discussion section, where we also say that “we expect that this bioeconomic approach will expand in the coming years due to the growing demand for bio-products and more specialized or niche foods.” We now added some specifications at the beginning of the concluding remarks.

In terms of cooperation, in the results we have shown how the different clusters are locally embedded and how they cooperate; we did not want to repeat this in the Concluding remarks.

Whether and to what extent the results/lessons learned are applicable in other developing / transition countries?

Answer: This question we cannot answer. Of course, one could say that in “similar” countries, with similar biomass and technology basis, maybe Brazil, the results are also applicable, but we are really reluctant to make such statements, as it would require to go into detail, which would require another study.

Policy implications

If one of the main aims is to contribute to the policy debate by demonstrating how bioeconomy ventures can differ due to the nature of economic activity and innovation strategies as well as the nature of local embeddedness, then one would also expect a discussion about more targeted policies regarding the different types of bioeconomy ventures (biomass and mass-production model, biotech-based model and bioembedded model).

Answer: You are right, a discussion on targeted policies would be very important, and we mention this now (p. 21). There might be some general policies for all types, and others, more specific policies. But to discuss which policies for which models, would go beyond the reach of our article, because then we would have to analyze which policies exist, how appropriate they are to promote the different models, which specific programs are needed, etc. Our more modest aim was to show that new models are emerging, which might need specific policies, but more research is needed here, as we stated, so that “appropriate support policies could be tailored to the needs of those ventures.”(p. 22)

In addition, the authors tend to overemphasize the relevance of bioembedded model (or agro-ecological model) in the transition processes, or at least, the role of these particular ventures is not sufficiently explained or supported by the data. E.g., see Table 4 and related explanations of how the innovativeness of this particular model is reflected merely via a focus on niche and specialized products? I.e., the reliance on biotechnologies or scientific cooperation does not seem to support the provision of innovative and premium level products in this cluster.

Answer: We agree that the relevance of the bioembedded model is still rather limited, but we do not see that in our article this relevance is overemphasized. We just say that in the future, it might be an interesting model for the bioeconomy in Argentina, as it might lead to “the construction of a new model of local productive development, more respectful of the environment and the circular economy”, because of the “concern for the environment, identity and the construction of local development processes…” (p. 16). These characteristics can be found in Tables 4 and 5. As to the innovativeness of this model, the bioembedded model has lower scientific cooperation, and uses less biotechnologies, but relies more on local and traditional knowledge, but also has patent use. It mainly follows pathway 5, creating new, innovative products with local value added, with ”rather low-tech innovations that utilize resources locally available, thereby adding value, bridging local products to non-local customers, and contributing to a more circular, sustainable economy.” (p. 19). We now made clear in the Conclusions that this approach is still has limited relevance (p. 21): “Even if this approach currently might have only limited relevance for the territorial development in Argentina, a significant contribution can be expected in the future and it can become an opportunity for development and territorial balance in a country deeply marked by territorial imbalances.”  

Specific comments

On page 9, lines 24-26, it is mentioned that four variables were decisive in the formation of three key models in the Argentine context, while on page 16, lines 28 and 29, it is claimed that these three models were formed based on only two variables?!

Answer: This relates to the simplified methods, not using cluster analysis, but an algorithm (described in the same section) that only used biomass and biotechnology as variables. As this algorithm produced a clearer distinction between the models, we used it instead of the cluster analysis with four variables. We now added a half-sentence to make this clearer (p. 10).

On page 7, lines 4-5, it is said that “It is not known how the lists exactly came about, …” – is this wording scientific enough?

Answer: You are right, it is too colloquial. What we wanted to express is that there is a lack of information, and a lack of transparency of how the existing information was created. We now say: “Since we do not know the criteria used to create the lists, nothing can be said about possible selection biases.”

Round 2

Reviewer 2 Report

I think the paper that was already strong from the beginning has improved and is ready for publishing. I really enjoyed the reading and look forward to read more papers on the bioeconmy from the authors. 

Author Response

Thank you again for your comments! We checked the whole document for English language and style and hope it has improved.

Reviewer 3 Report

The authors have made improvements in the draft and reacted to the main concerns. The case-specific narrative (how to understand the complexity of societal transformation towards sustainable bioeconomy in the developing country with a high biomass potential) is better explained, incl why the embeddness of a socio-ecological approach is important in this particular context.

Terminology needs to be double checked and its use made consistent. Look at the abstract in particular, but also in general. It is hard to track the specific aims of terms used such as "models" changed to "approaches", "approaches" changed to "typologies", "typologies" changed to "types" or vice versa "types" changed to "typologies" (for the latter, see the title of Table 1 and the (added) text linked to it)? What is the difference of types/groups/approaches/typologies, etc? Can You simplify here?

PS! On page 9 "live sciences" -> "life sciences".

Author Response

Thank you again for your comments! We checked again to make the terms consistent. We use “model” in a general term (“business model”, development model) or if other authors used the term in relation to the bioeconomy. We do not use “model” for our results, where we only use “approach(es)”. “Typology” refers to the intent of other authors to develop different types of the bioeconomy, as shown in Table 1. (Ideal) “types” are the outcome of such typologies, i.e., a systematic classification according to their common characteristics. We than do a Cluster analysis, and use “Cluster” or “groups” referring to the outcome of the analysis, which we then refer to the main three approaches, which are the outcome of the typologies shown in Table 1.

PS! On page 9 "live sciences" -> "life sciences".

Thanks, corrected.